# Dust impacts on Indian summer monsoon: chaotic or physical effect?

Jiawang Feng[1], Chun Zhao[1,2,3], Jun Gu[1], Gudongze Li[1], Mingyue Xu[1], Shengfu Lin[1], Jie Feng[4]

[1] Deep Space Exploration Laboratory/School of Earth and Space Sciences/CMA-USTC Laboratory of Fengyun Remote Sensing / State Key Laboratory of Fire Science / Institute of Advanced Interdisciplinary Research on High-Performance Computing Systems and Software, University of Science and Technology of China, Hefei, China

[2] Laoshan Laboratory, Qingdao, China

[3] CAS Center for Excellence in Comparative Planetology, University of Science and Technology of China, Hefei, China

[4] Department of Atmospheric and Oceanic Sciences and Institute of Atmospheric Sciences, Fudan University, Shanghai, China

*Correspondence to*: Chun Zhao (chunzhao@ustc.edu.cn)

**Abstract.** Aerosols have significant impacts on regional climate, which has been widely investigated with numerical experiments. However, uncertainties of simulated aerosol impact due to long-standing chaotic effect remain unclear. Here we propose a diagnostic method based on large ensemble simulations and random sampling algorithm to unveil the chaos-induced uncertainties in simulated aerosol climatic impacts that is overlooked in previous studies. Taking dust impacts on Indian summer monsoon system as a demonstration, our findings reveal that, while dust generally enhances the large-scale summer monsoon circulation consistently among ensemble members, its impacts on regional systems, such as monsoon depressions, exhibit significant chaotic effect: the simulated aerosol impacts on precipitation from individual ensemble member differ substantially, even inversely. Through quantitative analysis, we demonstrate that the magnitude of these chaotic effects diminishes following a $N^{-\frac{1}{2}}$ relationship with ensemble size N. Furthermore, our results indicate that statistical significance testing alone may be insufficient for robust attribution of dust impacts, as even small ensembles can yield statistically significant yet contradictory results. This study emphasizes the necessity of employing adequate ensemble sizes to capture reliable physical impacts of aerosol on regional climate.

## 1 Introduction

Aerosols, consisting of suspended solid and liquid particles in the atmosphere, play a crucial role in modulating both regional (Bollasina et al., 2011; Li et al., 2016) and global (Ramanathan et al., 2001; IPCC, 2014; Bellouin et al., 2020) climate systems through various pathways (Rosenfeld et al., 2007, 2008), mainly through aerosol-radiation interactions (ARI) and aerosol-cloud interactions (ACI) (IPCC, 2014). ARI involve scattering and absorption of radiation, thereby altering the Earth's radiation budget (Zhao et al., 2010, 2013a). ACI occur as aerosols serve as cloud condensation nuclei (CCN) and ice nuclei (IN), modifying cloud properties, precipitation patterns, and atmospheric dynamics (Fan et al., 2016; Ghan et al., 2016). These aerosol-induced modifications can significantly impact regional circulation patterns, precipitation distributions, and

temperature profiles, ultimately influencing climate variability on various temporal and spatial scales (Ramanathan et al., 2001; Rosenfeld et al., 2008; Zhao et al., 2011, 2012, 2020).

The complex nature of aerosol impacts has necessitated the development and application of sophisticated numerical models. These models have emerged as essential tools for understanding the complex impacts of aerosols on climate systems. Modern climate models can simulate the emission, transport, transformation, and removal of aerosols, along with their interactions with radiation and cloud processes  (e.g., Fast et al., 2016; Feng et al., 2023). However, significant uncertainties persist in numerical simulations of aerosol impacts. These uncertainties can arise from several sources such as limited model resolution affecting the representation of small-scale physical processes, simplified parameterizations of aerosol physical and chemical processes (Kinne et al., 2006; Zhao et al., 2013b), and incomplete understanding of aerosol-cloud-radiation interaction mechanisms (Zhao et al., 2011; Myhre et al., 2013; Ghan et al., 2016; Kok et al., 2023). Beyond these widely recognized sources of uncertainty, the inherent chaotic nature of the climate system may also lead to significant simulation uncertainties. However, research on how chaotic effects influence the simulation of aerosol climate impacts remains relatively limited.

In weather and climate research, the chaotic effects induced by initial condition perturbations have received widespread attention (e.g., Lorenz, 1963; Giorgi and Bi, 2000; Bei and Zhang, 2007; Hohenegger and Schar, 2007; Zhang et al., 2019; Judt, 2020). Since Lorenz (1963) first discovered weather systems' sensitive dependence on initial conditions, numerous studies have investigated the impact of this "butterfly effect" on weather forecasting and climate simulation.  For example, Giorgi and Bi, (2000) examined regional climate model sensitivity to initial conditions and found that the model internal variability significantly influences the day-to-day model solution, especially for summer precipitation: the domain-averaged daily precipitation RMSD was of the same order of magnitude as the average precipitation. Zhang et al., (2019) explored the influence of initial perturbations on predictability of weather forecasts in global climate models. (Hohenegger and Schar, 2007) demonstrated that cloud-resolving models are even more sensitive to initial perturbations than synoptic-scale models, with error growth rates about 10 times faster. Bei and Zhang, (2007) found that error growth is strongly nonlinear and small-amplitude initial errors, which are far smaller than those of current observational networks, may grow rapidly and quickly saturate at smaller scales. They subsequently grow upscale, leading to significant forecast uncertainties at increasingly larger scales. O'Brien et al., (2011) indicated that intrinsic variability (IV) of precipitation in regional climate models can be large enough to violate the assumptions of sensitivity study. These studies demonstrate that even negligible initial field perturbations can lead to significant differences in simulation results.

Nevertheless, currently, many studies rely on single numerical experiment to evaluate aerosol climate effects, potentially introducing significant uncertainties in interpreting modeling results (e.g., Wang et al., 2009; Zhong et al., 2017). Ensemble experiments, which involve running multiple simulations with slightly varying initial conditions or model parameters to capture a range of possible outcomes, have been widely employed to address these chaotic uncertainties (Bassett et al., 2020;

Laprise et al., 2012; Schellander-Gorgas et al., 2017; Feng et al., 2024b). While ensemble approaches have been widely adopted to address the uncertainties arising from chaotic effects, most studies utilize relatively small ensemble sizes of typically around 10 (e.g., Meehl et al., 2008; Vinoj et al., 2014; Jin et al., 2015; Solmon et al., 2015; Lau et al., 2017). Whether this limited ensemble size adequately characterizes the uncertainties introduced by chaotic effects remains unknown. Moreover, the quantitative characteristics of chaotic effects on aerosol-induced impacts on climate systems require further investigation.

To better understand the role of chaotic effects in simulating aerosol climate impacts, this study focuses on the Indian summer monsoon (ISM) region. This region exhibits high aerosol concentrations with complex spatiotemporal distributions and significant impacts on regional climate systems. Some studies have shown that aerosols influence Indian summer monsoon evolution through various mechanisms, including modification of radiation budgets, atmospheric thermal structure, and cloud microphysical processes (Lau et al., 2006, 2017; Lau, 2014, 2016; Sanap and Pandithurai, 2015). Despite significant progress, significantly different regional spatial and temporal details (even opposite results) have been found in many global or regional climate models (Jin et al., 2014; Vinoj et al., 2014; Jin et al., 2015; Solmon et al., 2015; Lau et al., 2017), indicating that there are still uncertainties remain in understanding aerosol impacts on the monsoon system in this region. For example, Vinoj et al. (2014) found rainfall increases mainly concentrated in southern India with minimal changes or decreases in central India; Jin et al. (2014) observed widespread rainfall enhancement across Pakistan and most of India, with maximum increases in the Indo-Gangetic Plain region; Solmon et al. (2015) reported yet another pattern, with increased rainfall in southern India but decreased precipitation in central and northern India and Pakistan. These divergent results making it an ideal case study for investigating the influence of chaotic effects in simulating aerosol climate impacts.

While substantial progress has been made in characterizing dust-monsoon interactions, most previous studies have focused on the mature monsoon season (July-August), during which atmospheric circulation is more stable and convective systems are already well established. In contrast, the onset phase is dynamically transitional and thus more sensitive to radiative and thermodynamic perturbations. During this transition, atmospheric circulation is dynamically unstable, the Intertropical Convergence Zone(ITCZ) and low-level jets are reorganizing, and synoptic systems such as monsoon depressions are forming. Under such complex conditions, dust-induced heating may exert outsized influence. Furthermore, to investigate the influence of chaotic effects of dust impacts, we plan to conduct a large ensemble of experiments with 50 members, which demands substantial computational resources. Given that dust may exert a pronounced influence during the onset period and to manage the computational resource constraints, we select only the onset period of the ISM in 2016 (June 10–30) as our simulation period.

This study has three primary objectives: (1) to quantify the uncertainties in simulating aerosol impacts introduced by chaotic effects, (2) to distinguish between physical and chaotic effects in the dust aerosol impacts on ISM system, and (3) to determine whether simulated aerosol impacts on the ISM are predominantly driven by physical processes or significantly influenced by

chaotic behaviors. We define the "physical effect" as the deterministic response of meteorological fields to aerosols that remains consistent across ensemble members despite initial condition perturbations. The ensemble-mean approximates this underlying physical effect by averaging out chaotic influences. Conversely, the "chaotic effect" represents internally generated variations arising from initial condition perturbations, manifested as the spread among ensemble members (Feng et al., 2024a).

The remainder of this paper is structured as follows: Section 2 describes our methodology, including the iAMAS model employed (Section 2.1), experiments configurations and methods for generating perturbed initial conditions (Section 2.2), and observational datasets used for validation (Section 2.3). Section 3 presents our analysis of chaotic effects on dust aerosol impacts on the ISM and discusses the relationship between ensemble size and chaotic uncertainties. Section 4 provides conclusions and summarizes the implications of our findings and discusses the limitations of this study.

## 2 Methods

### 2.1 Model

In this study, we employed the integrated Atmospheric Model Across Scales (iAMAS) (Feng et al., 2023; Gu et al., 2022). The iAMAS model is a non-hydrostatic global variable-resolution atmospheric modeling system featuring online integrated aerosol feedbacks. The model is also designed for the supercomputer with heterogeneous many-core architecture such as China's Sunway supercomputer.

iAMAS's dynamic core is adapted from the Model for Prediction Across Scales – Atmosphere (MPAS-A) (Skamarock et al., 2012), which discretizes the computational domain horizontally on a C-grid staggered unstructured Voronoi mesh using finite-volume formation (Skamarock et al., 2012). The fully compressible non-hydrostatic equations are casted in terms of geometric-height hybrid terrain-following coordinate, and the solver applies the split-explicit time integration scheme. The time-integration scheme employs the 3rd-order Runge-Kutta (RK3) method and the explicit time-splitting technique (Wicker and Skamarock, 2002).

For physics suite, iAMAS incorporates a comprehensive suite of microphysical parameterization schemes, including the Predicted Particle Properties (P3) scheme (Morrison and Milbrandt, 2015), the Morrison double-moment scheme (Morrison et al., 2005), and the Thompson scheme (Thompson et al., 2008) , the WRF Single-Moment 6-class scheme (WSM6) (Hong and Lim, 2006), and the basic warm-rain Kessler scheme (Kessler, 1969). On convective processes, iAMAS implements multiple parameterization options: the sophisticated multi-scale Kain-Fritsch (MSKF) scheme (Zheng et al., 2016), the original Kain-Fritsch (KF) scheme (Kain, 2004), the original and new Tiedtke mass-flux schemes (Tiedtke, 1989; Zhang et al., 2011), and the modified version of the scale-aware Grell-Freitas scheme (Grell and Freitas, 2014). The surface layer physics options include the classical Monin-Obukhov similarity theory scheme (Monin and Obukhov, 2009) and the Mellor-Yamada-

Nakanishi-Niino (MYNN) scheme (Nakanishi and Niino, 2006, 2009). For planetary boundary layer (PBL) processes, both the Yonsei University (YSU) scheme (Hong et al., 2006) and MYNN scheme are implemented. The land-atmosphere interactions are represented through the Noah land surface model with four soil layers (Chen and Dudhia, 2001). Radiative transfer processes are parameterized using either the Rapid Radiative Transfer Model for GCMs (RRTMG) for both shortwave
and longwave radiation (Iacono et al., 2000; Mlawer et al., 1997) or the Community Atmosphere Model (CAM) radiation scheme.

For aerosol related suite, iAMAS includes the processes of online emission, advection, diffusion, vertical turbulent mixing, dry deposition, gravitational settling, and wet scavenging. In the experiments conducted for this study, only dust aerosols are included to isolate their effects from those of other aerosols. iAMAS uses sectional approach to represent a 10-bin size
distribution of aerosol particles ranging from ~0.04 to 40 μm. Each size bin is assumed to be internally mixed so that all particles within a size bin have the same properties. The dust emission scheme of iAMAS is adapted from the Goddard Chemistry Aerosol Radiation and Transport (GOCART) scheme (Ginoux et al., 2001). The dry deposition of aerosols is calculated based on Peters and Eiden, (1992) in iAMAS and wet deposition of aerosols both in-cloud and below-cloud are also treated in the model.

Aerosol-cloud interaction (ACI) is implemented in the model based on the method described by (Gustafson et al., 2007) for calculating the activation and resuspension between dry aerosols and cloud droplets. Aerosol activation (or droplet nucleation) is based on a maximum supersaturation determined from a Gaussian spectrum of updraft velocities, similar to the methodology used in (Ghan et al., 2001). The activated droplet number is then coupled with the Thompson microphysics scheme. In this way, aerosols can affect cloud droplet number, and clouds can also alter aerosol concentration through aqueous processes and
wet scavenging. The hygroscopicity of dust aerosols are assumed to be 0.10 in this study. Within the Thompson cloud microphysics scheme, the number of ice nucleation (IN) in mixing-phase clouds from dust is calculated following the formula proposed by DeMott et al.(DeMott et al., 2010). This study only considers the wet scavenging process of activated dust aerosols into cloud droplet, ignoring the conversion of dust into IN because the IN feedback calculations are not fully evaluated in iAMAS at this stage.

iAMAS also incorporates the aerosol-radiation interaction (ARI). Following the new method proposed by Feng et al., (2025), aerosol optical properties are computed and coupled with the RRTMG radiation scheme for both shortwave and longwave bands. For dust aerosols, this study utilizes the Optical Properties of Aerosols and Clouds (OPAC) dataset (Hess et al., 1998) to provide their shortwave and longwave refractive indices.

Recent studies have demonstrated the diverse capabilities of iAMAS. Feng et al., (2023) implemented an aerosol modeling
framework into iAMAS along with simulations of aerosol-radiation and aerosol-cloud interactions. Their study evaluated the

model's capability in simulating atmospheric dust and examined how mesh refinement impacts dust simulations. Gu et al., (2022) achieved significant improvements in computational efficiency and reduced input/output (I/O) costs through multi-dimension-parallelism structuring, aggressive and finer-grained optimization, manual vectorization, and parallelized I/O fragmentation. These enhancements achieved the speed of 0.82 simulation day per hour with forecasts including online aerosol simulations at a global convection-permitting scale with 3km resolution. In a subsequent study, Gu et al., (2024a) conducted comparative one-month forecasts at different resolutions (global 3km, variable 4-60 km, and global 60 km) employing iAMAS. Their results revealed that the global 3km resolution forecast accurately captured the plum rain rainband around Japan, while lower-resolution forecasts showed northward displacement and weaker intensity, attributed to shifted atmospheric rivers over Japan. Gu et al., (2024b) employed iAMAS at a 3-km resolution and achieved unprecedented accuracy, reducing track errors to below 100 km over a 120-hour forecast period. Notably, iAMAS successfully predicted Typhoon In-fa's sudden track changes and dual landfall locations, outperforming current operational forecasts. Li et al., (2024) carried out global simulations with uniform resolution, and found that high spatial resolution (global 3km) experiment suppresses the excessive equatorial light rain simulated by experiment at coarser resolution (global 60km) and improve the dry bias of the South Asia summer monsoon rainfall over northern India by modulating the competition between the maritime and continental rainfall band. The successful applications of iAMAS across diverse research contexts have demonstrated its capability and reliability. The model's feature of global variable-resolution mesh makes it suitable for future high-resolution studies of local aerosol-climate impacts, while simultaneously enabling the investigation of cross-scale interactions between aerosols and climate systems from regional to large scales.

## 2.2 Numerical Experiments

### 2.2.1 Configuration of simulations

We conducted two sets of ensemble experiments, each comprising 50 members with perturbed initial conditions generated using the method detailed in Sect. 2.2.2. The first set, called the "Control" experiment, included simulations with dust aerosols, while the second set, termed the "Sensitive" experiment, excluded dust aerosol emissions to examine their impacts on the ISM system. To isolate the influence of the local dust, the "Sensitive" experiment specifically eliminated dust aerosol emissions only in the Arabian region (7.5N~42N, 31E~78E, marked in Fig.  S1 in supplement materials), while maintaining all other settings identical to the "Control" experiment.

The simulations covered the period from June 10 to June 30, 2016, focusing on a specific intense rainfall period occurring during the 2016 Indian summer monsoon season. To be clarified, this period does not cover the entire dust-ISM interactions throughout the monsoon season or across different years. We selected this specific period as it features a monsoon onset period with monsoon depression system that is particularly sensitive to aerosol impacts, making it suitable for investigating physical and chaotic effects. This approach also balances computational costs (necessitated by the large number of ensemble

experiments) with scientific objectives, though we recognize that longer-term simulations would be valuable for future work to capture the full range of dust-ISM interaction. We employed a quasi-uniform mesh with approximately 60 km grid spacing. The model's top height is set at 30 km, with 55 vertical layers. Initial meteorological conditions were derived from the European Centre for Medium-Range Weather Forecasts (ECMWF) ERA5 reanalysis dataset (Hersbach et al., 2020) , utilizing data at 0.25° horizontal resolution and 6-hour temporal intervals. Sea surface temperatures, prescribed from the ERA5 reanalysis dataset, were updated every 6 hours throughout the simulation period. This approach is common for short-term atmospheric process studies as the simulation period (20 days) is short compared to typical SST adjustment timescales. Besides, since SST is prescribed, the model differences will only be attributed to dust aerosol effects associated with aerosol-monsoon interaction. The model physics configuration incorporated several well-established schemes: the Mellor-Yamada-Nakanishi-Niino PBL scheme (Nakanishi and Niino, 2006, 2009), the NOAH land-surface scheme (Chen and Dudhia, 2001), the Thompson microphysics scheme (Thompson et al., 2008), the Grell-Freitas cumulus convection scheme (Grell and Freitas, 2014), and the RRTMG longwave and shortwave radiation schemes (Iacono et al., 2000; Mlawer et al., 1997). The dust simulation framework followed the methodology established by Feng et al., (2023).

## 2.2.2 Generating Perturbed Initial Conditions for Ensembles

In this study, we employed the Breeding of Growing Modes (BGM) technique to generate initial perturbed conditions in ensemble simulations. The BGM method is straightforward to implement, computationally efficient, and superior in sampling physically-balanced spatial uncertainties in initial conditions. This technique was first introduced by the National Centers for Environmental Prediction (NCEP) for generating initial perturbations of the global ensemble forecast system (Toth and Kalnay, 1993, 1997). The BGM method effectively captures the fast-growing perturbation directions conditioned on the reference background states with very low computational cost. This advantage of these initial ensemble perturbations favors the diverse evolution of perturbed simulations, enhancing the reliability of model ensembles.

Based on the original BGM method, we also adapt it for use with SCVTs (Spherical Centroidal Voronoi Tessellations) grid characteristics to generate unstructured perturbation initial fields for iAMAS. The perturbation amplitude is calculated based on moist energy norm, with perturbations applied to three initial variables: potential temperature, surface pressure, and specific humidity. The calculation of the moist energy norm follows the method outlined by Ehrendorfer et al., (1999):

$$\frac{1}{2}\frac{1}{D}\int_D \int_0^1 \left[ u'^2 + v'^2 + \frac{c_p}{T_r}T'^2 + RT_r\left(\frac{p'}{p_r}\right)^2 + \epsilon \frac{L^2}{c_p T_r}q'^2 \right] d\sigma dD \tag{1}$$

where $c_p$ (1005.7 J kg$^{-1}$ K$^{-1}$) is the specific heat capacity at constant pressure, $R$ (287.04 J kg$^{-1}$ K$^{-1}$) is the gas constant for dry air, $L$ (2.5104 × 10$^6$ J kg$^{-1}$) represents the latent heat of vaporization, $T_r$(270 K) denotes the reference temperature, and

$p_r$ (1000 hPa) is the reference pressure. The terms $u'$ and $v'$ represent the differences between the simulation results and reanalysis fields of zonal and meridional wind components, respectively. $T'$ denotes the temperature difference, $p_s'$ represents the surface pressure difference, and $q'$ indicates the specific humidity difference. $D$ corresponds to the model grid cell area, and $\epsilon$ is the normalization factor for specific humidity, which is set to unity in this study.

The procedure for generating initial perturbation fields on SCVTs grids is as follows:

1. **Initialization of Perturbations**: A 24-hour simulation is conducted 48 hours in advance. The initial perturbation amplitude (i.e., moist energy norm) is computed by comparing the simulation results with reanalysis data. Subsequently, each grid point's root mean square error (RMSE) with its neighboring model cells is calculated and multiplied by a random number to generate the initial perturbations. The RMSE is calculated as: $RMSE_{Cell_a} = \sqrt{\frac{\sum_{i=1}^{n}(M_{Cell_i} - R_{Cell_i})^2}{n}}$, where $M$ represents model values, $R$ represents reanalysis values, and $n$ is the number of

neighboring grid points plus one, including $Cell_a$ itself.

    2. **Breeding Cycle**: Every 6 hours, the simulation results are compared with reanalysis data to compute the scaling factor by comparing the moist energy norm with the initial perturbations. This scaling factor is then applied to the difference between the model output and the reanalysis data. The scaled perturbations are added to the reanalysis field to replace the corresponding model variables, and the breeding cycle is continued.

3. **Mature Perturbations**: Based on previous study (Toth and Kalnay, 1997), perturbations typically mature after 48 hours of breeding. These perturbation fields are then superposed on the reanalysis state to produce initial members for ensemble simulations.

    4. **Repetition**: Different random seeds are used for each initialization to generate other perturbated initial conditions as ensemble members.

The spatial distributions of initial surface potential temperature, surface pressure, and surface specific humidity across the 50 ensemble members over the Indian monsoon region are presented in Fig. S2-S4 (supplementary materials). The perturbations introduced in these initial conditions exhibit minimal magnitude. To quantify these subtle perturbations, we calculated the deviations of individual members from the ensemble mean (Fig. S5-S7). Despite the small magnitude of initial perturbations, these deviations reveal random variations among ensemble members, confirming the effective implementation of our

perturbation methodology in generating perturbed initial conditions while maintaining physical consistency within the meteorological fields.

## 2.3 Datasets

To evaluate the model performance and validate our simulation results, we utilized multiple observational and reanalysis datasets. The Multi-angle Imaging Spectro Radiometer (MISR) aboard NASA's Terra satellite provides global aerosol optical depth (AOD) measurements (Diner et al., 1998). We employed the MISR Level 3 version F08_0031 daily aerosol product with a spatial resolution of 0.5° × 0.5°. MISR's unique multi-angle observation capability enables accurate aerosol retrievals over both land and ocean surfaces, making it particularly suitable for monitoring dust aerosols over the ISM region.

To validate circulation conditions of the atmosphere, we used the fifth-generation ECMWF reanalysis (ERA5) dataset. ERA5 provides high-resolution global analyses of atmospheric parameters at 0.25° × 0.25° spatial resolution and hourly temporal resolution with 37 vertical levels. The dataset incorporates various observation systems and advanced data assimilation techniques, offering reliable representations of atmospheric states (Hersbach et al., 2020).

Precipitation data were obtained from the Climate Prediction Center Morphing Technique (CMORPH) Version 1.0 dataset, which provides global precipitation estimates at high spatial (0.25° × 0.25°) resolutions (Joyce et al., 2004).This dataset is particularly valuable for analyzing precipitation patterns over the Indian monsoon region due to its consistent spatial and temporal coverage.

## 3 Results

### 3.1 Chaotic effects on simulated Indian Summer Monsoon

Figure 1 illustrates the comparative analysis between observational data and numerical simulations of monsoon circulation (represented by 850 hPa wind fields) and precipitation patterns during the monsoon onset period (June 10-30, 2016). The observed wind field at 850hPa is derived from ERA5 and the observed rainfall is from CMORPH. The 850 hPa wind field from ERA5 reveals key features of the early-summer monsoon circulation: a well-established cross-equatorial flow over the Arabian Sea that develops into strong southwesterly winds along the western Indian coast. This low-level jet serves as the primary moisture transport pathway. The precipitation distribution from CMORPH during this period shows three major rainfall zones: an intense precipitation band along the Western Ghats due to orographic lifting of moisture-laden monsoon winds, a broad rainfall maximum over the Bay of Bengal, and substantial precipitation over the northern Indian subcontinent where monsoon depressions frequently occur (Li et al., 2016; Srivastava et al., 2017). The ensemble-mean results (Fig. 1b) are able to reproduce these fundamental features of the monsoon system. While the simulated intense precipitation zone along the western coast of India shows a southward displacement, the model effectively captures the overall spatial distribution of rainfall

and the large-scale circulation patterns, particularly the strong southwesterly monsoon flow and the precipitation associated

with monsoon depressions over northern India.

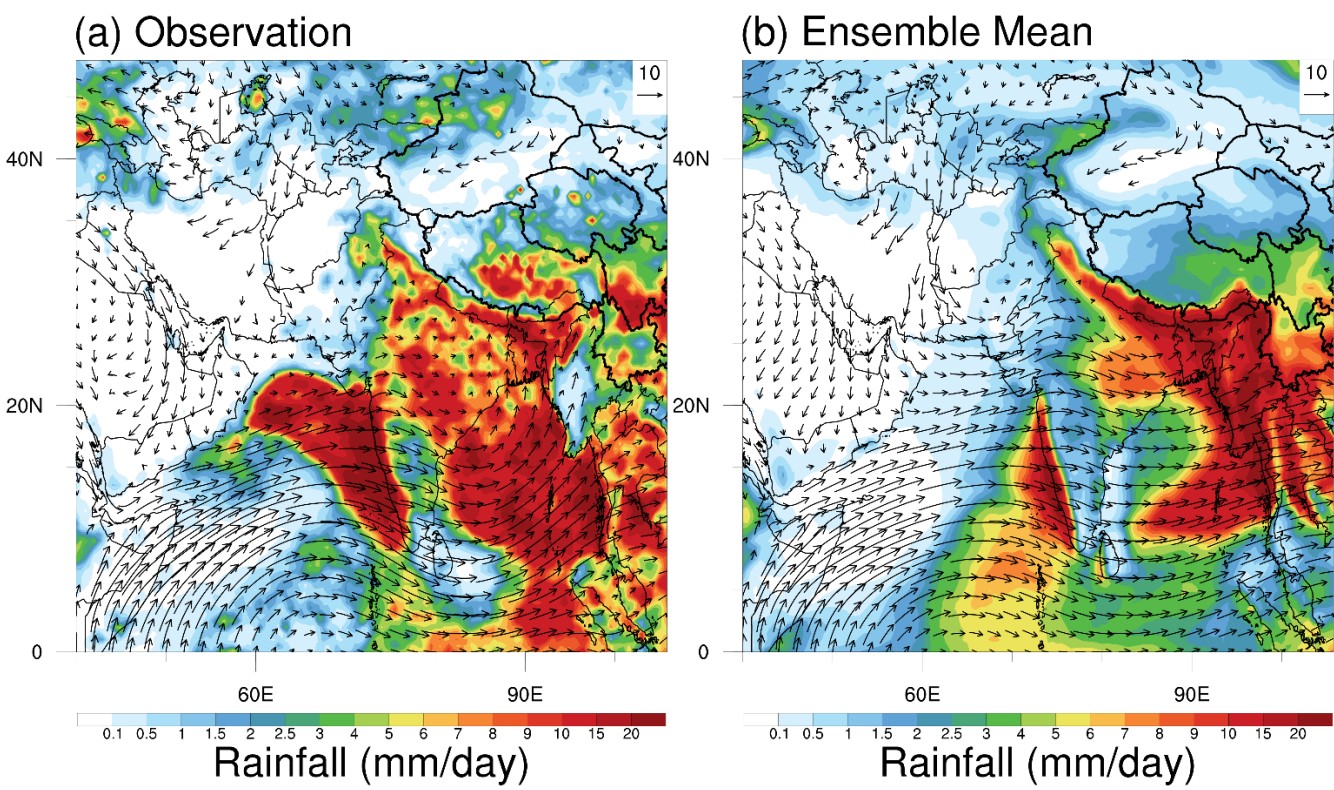

**Figure 1: The observations and simulations of monsoon circulation (wind field at 850hPa) and precipitation, averaged from June 10,**
**2016, to June 30, 2016. The observed wind field at 850hPa derived from ERA5 and the observed rainfall is from CMORPH. The**
**simulation results are shown as 50-member ensemble mean.**

As previously introduced, slight perturbations in initial conditions among ensemble members can lead to substantial

divergences in simulation outcomes. This sensitivity to initial conditions warrants a detailed examination of chaotic behavior

within these ensemble simulations. Figure 2 presents the precipitation patterns from 50 ensemble members of "Control"

experiments over the Indian monsoon region (the results of "Sensitive" experiments are illustrated in Fig. S8). While these

simulations exhibit some consistent features, such as notably the intense precipitation along the Himalayan southern slopes

and southwestern Indian coast, they demonstrate remarkable inter-member variability, particularly over northern regions of

the Indian subcontinent (highlighted by the black box in Fig. 2). Analysis of individual ensemble members reveals substantial

variation in their ability to simulate monsoon depression-associated precipitation. Several members (e.g., members 3, 9, 14, 17, 28, and 48) successfully capture the distinctive precipitation pattern associated with monsoon depressions. However, a subset of members (notably members 6, 18, 20, 30, and 49) fails to reproduce the precipitation in this region, highlighting the chaotic effect in simulating such synoptic-scale features.

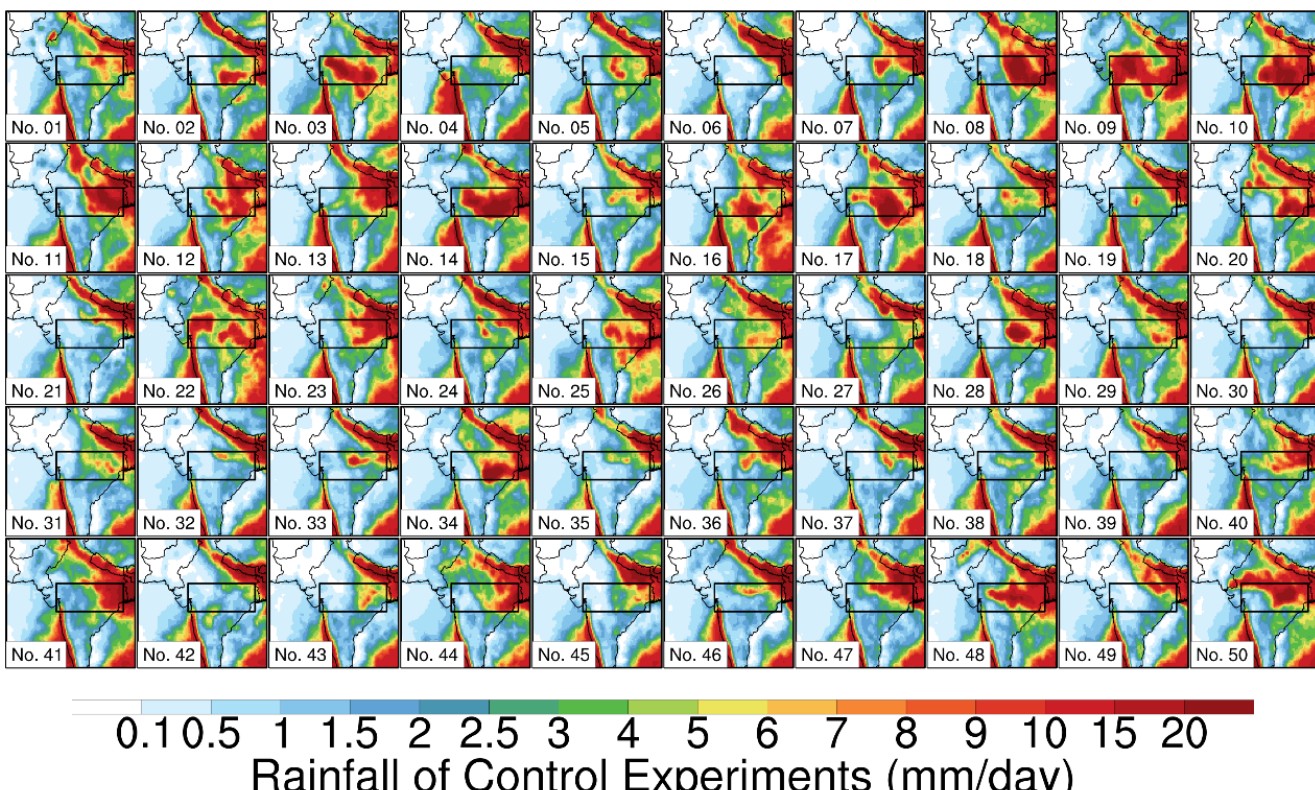

**Figure 2: The spatial distributions of precipitation derived from 50 ensemble members of "Control" experiments over the Indian monsoon region. The results are averaged from June 10, 2016, to June 30, 2016. The monsoon depression region is delineated by the black box.**

## 3.2 Chaotic effects on simulated dust aerosol impacts on ISM

Given that this study aims to investigate the impacts of dust aerosols on ISM precipitation, accurate simulation of dust concentrations serves as a fundamental prerequisite. Figure 3 presents a comparison of Aerosol Optical Depth (AOD) at 550 nm between satellite observations from MISR and the 50-member ensemble means for both control and sensitivity experiments. To ensure robust comparison with MISR observations, which are acquired from the Terra platform with an equatorial crossing time of approximately 10:45 local time, the model-simulated AOD values were temporally sampled to match Terra's overpass time. Additionally, spatial collocation was performed to align model outputs with MISR's valid retrieval grids.

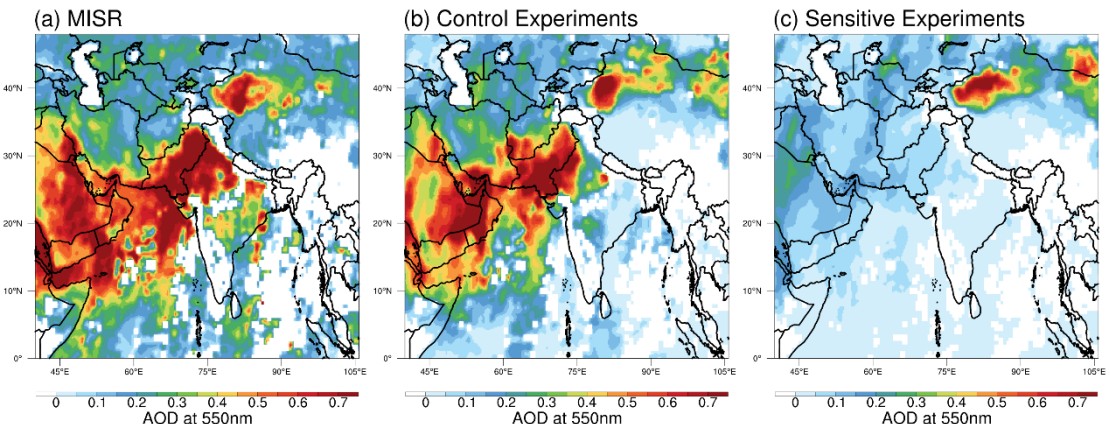

Figure 3: Spatial distribution of AOD at 550nm derived from (a) MISR; (b) Control experiments; (c) Sensitive experiments.

The spatial pattern reveals three distinctive high-AOD regions: the Arabian Peninsula, serving as the primary dust source region, the Arabian Sea, showing elevated AOD values due to dust transport along the monsoon flow path, and the Indo-Gangetic Plain, where AOD is from both transported and local dust. The monsoon precipitation regions show notably lower AOD values due to efficient wet removal processes. The control experiment's ensemble mean (Fig. 3b) successfully reproduces these observed AOD patterns, particularly the high values over the dust source and transport regions. In contrast, the sensitivity experiment (Fig. 3c), with eliminated Arabian dust emissions, shows significantly reduced AOD values, clearly demonstrating the dominant contribution of Arabian dust to the ISM aerosol loading.

Figure 4 illustrates the inter-member variability of AOD across 50 ensemble members. Notably, while the dust source and transport regions (Arabian Peninsula and Arabian Sea) show highly consistent AOD patterns, the monsoon precipitation

regions exhibit more noticeable inter-member variations. This spatial difference in ensemble spread suggests that in high-dust regions, the consistent AOD patterns indicate that dust's physical impacts on monsoon circulation should be similar across members, and in precipitation-dominated regions, the chaotic effects on AOD likely result from differences in wet removal processes among members, reflecting precipitation's impact on dust distribution rather than vice versa.

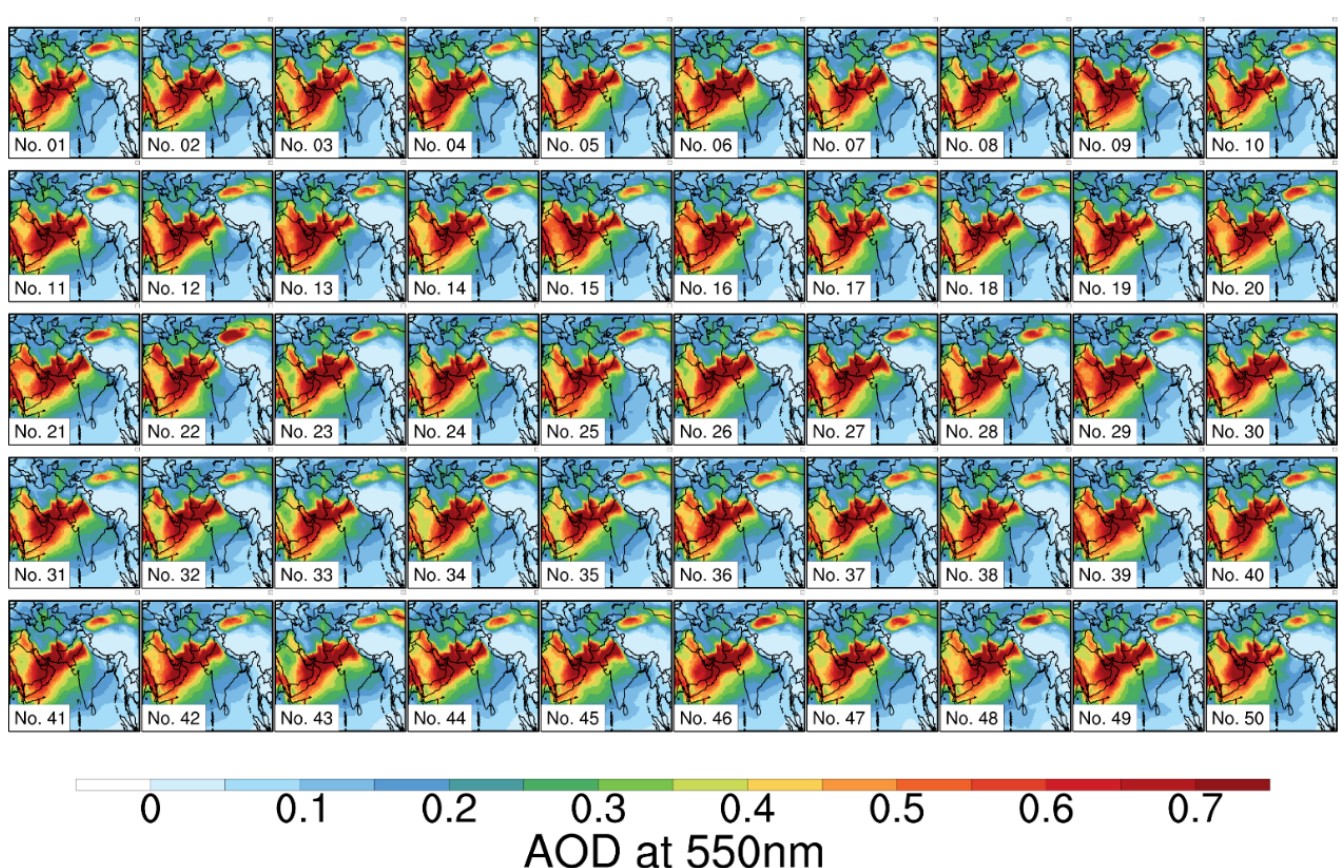

**Figure 4: Same as Fig. 2 but for AOD at 550nm of "Control" experiments.**

Figure 5 illustrates the ensemble-mean impacts of dust aerosols on the ISM rainfall and the wind field at 850hPa. The dust-induced impacts are derived from the difference between Control and Sensitivity experiments. The results reveal that dust

aerosols strengthen the southwesterly monsoon flow from the Arabian Sea toward the Indian subcontinent (as shown in Fig. 5). This response aligns with the basic dust-monsoon interaction mechanism proposed by Vinoj et al. (2014), where increased

atmospheric warming (see Fig. S9b) from high dust concentrations leads to a reduction in surface pressure and strengthening of the pressure gradient over the Arabian Sea. This pressure gradient enhancement drives stronger monsoon flow and moisture convergence. This enhanced circulation pattern leads to precipitation intensification along two primary regions: the western Indian coast and the western Himalayan foothills. Furthermore, dust aerosols generate a cyclonic wind anomaly in the monsoon depression region (delineated by the black box), consequently intensifying precipitation within this domain. The region-average precipitation in the "Control" experiment (5.27 mm/day) is nearly 100% higher than the "Sensitive" run (2.66 mm/day), revealing significant dust impacts on precipitation based on ensemble means results. The large magnitude of this dust-induced precipitation change can be attributed to the specific meteorological mechanism we investigated: dust aerosols' influence on monsoon during the monsoon onset. As we discussed in our analysis of individual ensemble members in Section 3.1, dust plays a role in determining whether monsoon depression-associated precipitation patterns develop successfully in our simulations. This binary-like behavior—where dust presence can influence whether or not a monsoon depression system forms—explains the large precipitation difference we observe. Monsoon depressions are known to produce large amounts of rainfalls, capable of generating several mm/day of precipitation over extensive areas (Srivastava et al., 2017). Therefore, the difference between successfully simulating versus missing such a system naturally leads to substantial percentage changes in regional precipitation. To be clarified, our results on precipitation response patterns reflect this specific meteorological situation (Jun 10 to Jun 30, 2016), and the large effect we document here specifically applies to dust's role during the monsoon onset period in modulating the formation of monsoon depression systems during favorable meteorological conditions, rather than representing a general dust-monsoon interaction magnitude that could be extrapolated to seasonal or climatological time scales.

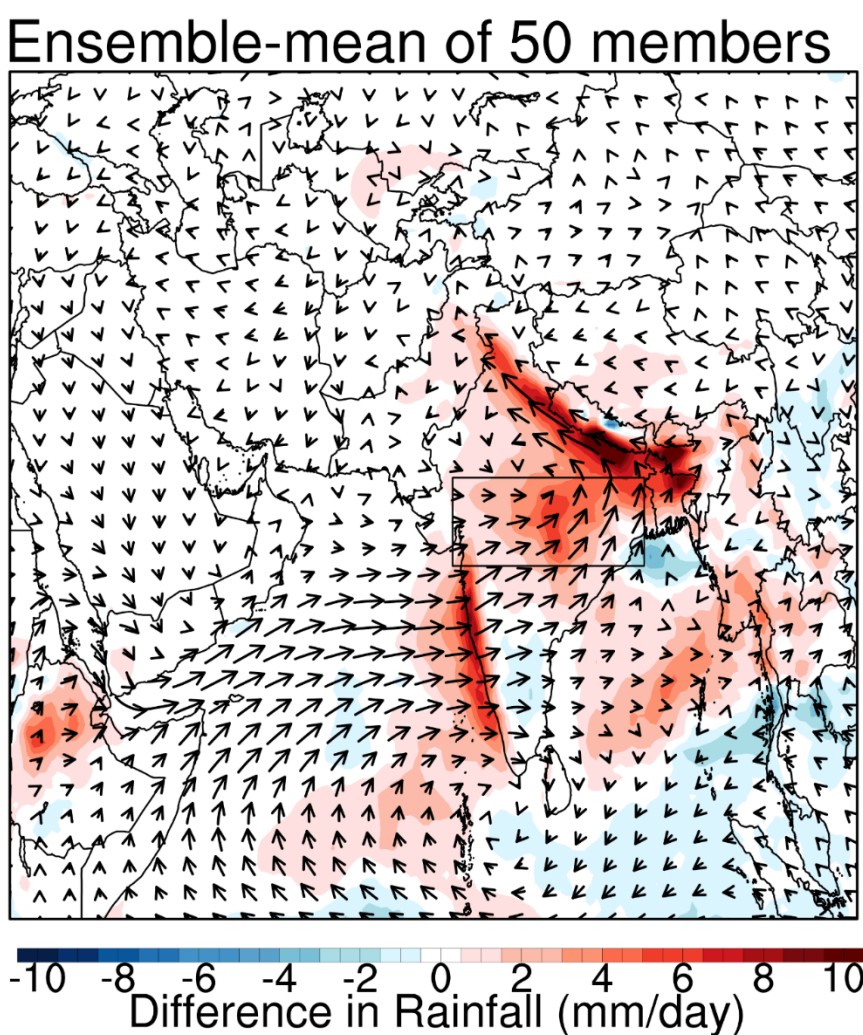

**Figure 5.** The ensemble-mean impacts of dust aerosols on the ISM rainfall and the wind field at 850hPa, represented by the differences in simulated results between the "Control" and "Sensitive" experiments.

Figure 6 presents precipitation and wind field responses across all 50 ensemble members, revealing both consistent signals and substantial inter-member variability in dust-induced impacts. Notably consistent features appear along the western coast of India and the Himalayan foothills, where most ensemble members show enhanced precipitation and strengthened southwesterly flows, suggesting these regions are less susceptible to chaotic effects.

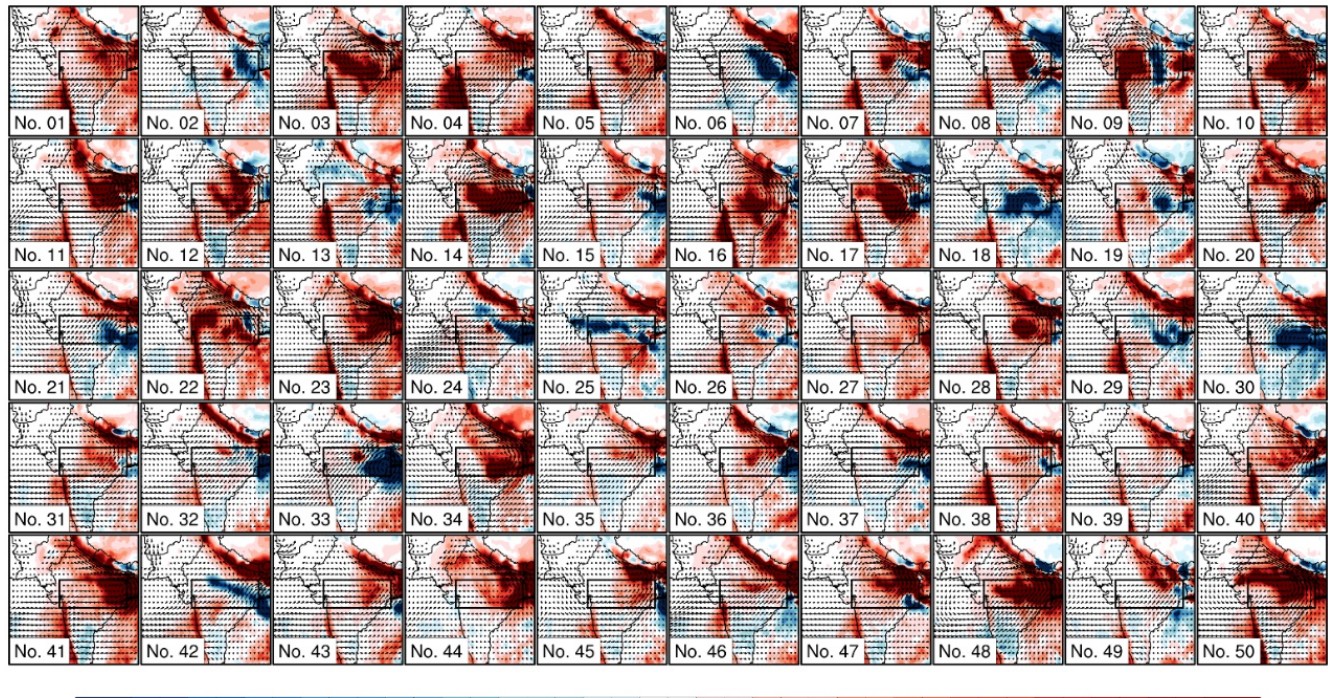

**Figure 6: Same as Fig. 5 but for each ensemble members.**

However, substantial chaotic effects emerge in specific regions, particularly within the monsoon depression region and central India, where ensemble members can yield opposing signs of precipitation response. This spatial pattern of uncertainty closely corresponds to regions that exhibit high chaotic effects in both "Control" (Fig. 2) and "Sensitive" (Fig. S8) simulations, indicating that areas naturally prone to chaotic behavior also show enhanced sensitivity to dust perturbations. This regional variability may help explain the contradictory findings in previous studies (Jin et al., 2014; Solmon et al., 2015; Vinoj et al., 2014; see Sect. 1 for a detailed discussion of these discrepancies). These divergent results, particularly in regions we identify as highly sensitive to chaotic effects, suggest that limited ensemble sizes in previous studies may have captured different dust-monsoon interaction, leading to contradictory conclusions about dust impacts in these regions.

### 3.3 Dependence of chaotic uncertainties on ensemble size

Our analysis demonstrates that dust aerosol impacts on regional-scale weather systems, particularly within the Indian Summer Monsoon region, display significant sensitivity to initial condition perturbations. This sensitivity manifests most prominently in the simulation of mesoscale features, where different ensemble members can produce opposing conclusions regarding dust impacts on precipitation and circulation patterns. Such divergent results could potentially lead to mischaracterization of dust-climate interactions if based on single simulations or limited ensemble sizes.

To address this challenge, we investigate the relationship between simulated uncertainty of dust impacts and ensemble size, focusing specifically on the monsoon depression region where the inter-member variability is most pronounced. We quantify the uncertainty through the analysis of dust-induced precipitation responses, calculated as the difference in area-averaged precipitation between "Control" and "Sensitive" experiments across our 50-member ensemble set within the monsoon depression domain (delineated by the black box in Fig. 5). Then we performed resampling without replacement on all these differences. This process involves selecting N (N represents the number of members in ensemble) difference values from the original dataset as a single sample. Consequently, the number of possible samples of size N is given by $C_{50}^{N} = \frac{N!}{N!(50-N)!}$. However, as N increases, the number of samples becomes exceedingly large (for example, $C_{50}^{15} \approx 2.25 \times 10^{12}$), making it impractical to calculate and analyze. Therefore, when the number of samples exceeds 10,000, we randomly select 10,000 samples for analysis. To quantify the uncertainties among members of ensemble, we calculated the 2.5th and 97.5th percentiles of the sample estimations' distribution. The range between these two percentiles is defined as the 95% confidence interval for N. In other words, there is a 95% probability that the result of a conducted ensemble simulation with N members falls within this interval. The narrower the confidence interval, the more reliable the ensemble simulation results with N members.

Figure 7 examines how ensemble size affects the uncertainty in estimating dust impacts on precipitation. Figure 7a shows the 95% confidence interval of regional average dust impacts on monsoon precipitation as a function of ensemble size, with the orange shading representing the spread of possible values. For small ensemble sizes (< 10 members), the distribution exhibits a wide spread, with some estimates even showing opposite signs of dust impacts. As the ensemble size increases, this spread gradually narrows, forming a more concentrated distribution around the mean value of approximately 2.8 mm/day (indicated by the white dashed line, representing the 50-member mean results). Figure 7a illustrates that, with smaller sample sizes, the impact of chaotic effects is significant, leading to a more dispersed distribution of the sample mean impacts of dust aerosols. The differences in precipitation among samples can even show opposite signs. This suggests that with a small ensemble size, the relationship between dust aerosols and the ISM precipitation appears highly chaotic, resulting in low reliability of the simulated conclusions. Fortunately, as the number of ensemble members increases, the sample estimates converge towards the true value (here, the average value of the 50 experiments is considered as the true value).

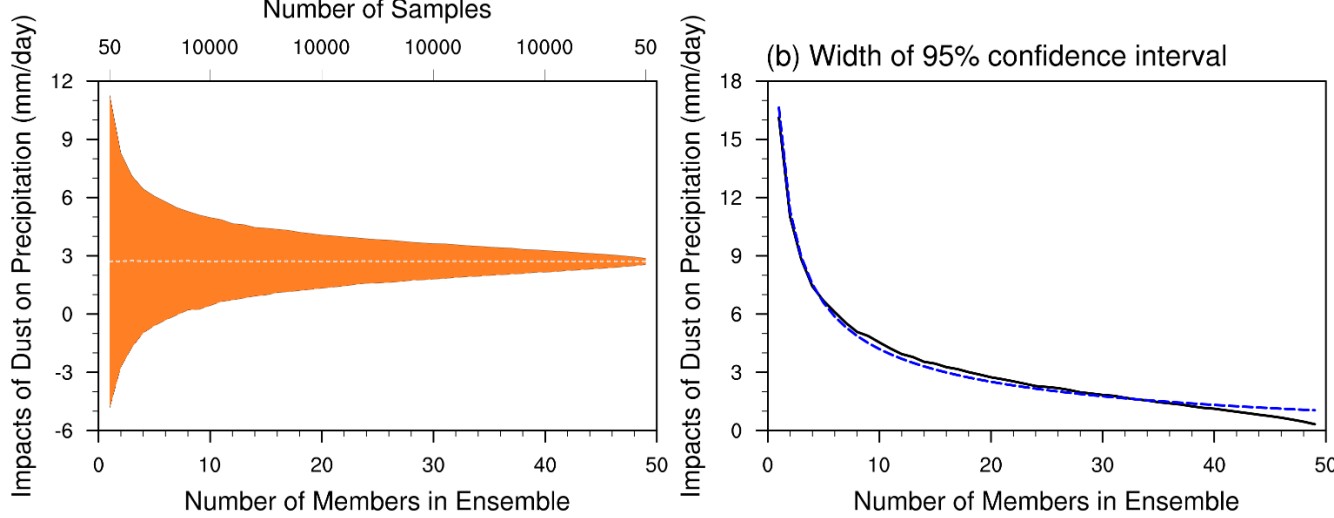

**Figure 7: 95% confidence interval of regional average precipitation difference. The vertical axis represents the difference in average precipitation between the control experiment and the sensitivity experiment within the monsoon depression area. The definition of the 95% confidence interval is the range between the 97.5th percentile and the 2.5th percentile of the sample. (a) Shows the 95% confidence interval, with the black dashed line representing the average precipitation difference of 50 experiments, and the white dashed line representing the average value for each number of members in ensemble. (b) The black solid line in the figure shows how the width of the 95% confidence interval varies with number of members in ensemble, and the blue dashed line represents the logarithmic fitting curve, with the fitting expression being $18.18N^{-\frac{1}{2}} - 1.55$.**

Figure 7b quantifies this uncertainty reduction by plotting the width of the 95% confidence interval against ensemble size. The confidence interval width decreases sharply from about ~16 mm/day with very few members to around 5 mm/day with 10 members, followed by a more gradual decline until with 50 members. The fitting results of Fig. 10b demonstrate that the width of the confidence interval is roughly proportional to $N^{-\frac{1}{2}}$, with the fitting expression being $18.18N^{-\frac{1}{2}} - 1.55$ for this case (see also O'Brien et al. (2011) for similar $N^{-\frac{1}{2}}$ convergence behavior with ensemble size)". In summary, increasing the sample size reduces the uncertainties of impacts of aerosols in ensemble simulations. However, as the number of members increases, the "cost-effectiveness" of further increasing the ensemble simulation size decreases.

While some studies have employed ensemble approaches to address uncertainties in aerosol-climate interactions (Meehl et al., 2008; Jin et al., 2015; Solmon et al., 2015; Lau et al., 2017), most researches utilize around or less than 10 ensemble members. Figure 8 illustrates the spatial distribution of dust-induced precipitation impacts for two extreme cases selected from 10,000 possible combinations of 10-member ensembles, representing the maximum (E1) and minimum (E2) area-averaged responses.

Both cases maintain certain common features, such as positive precipitation changes along the Himalayan foothills and enhanced westerlies over the Arabian Sea. However, they differ substantially in the magnitude and spatial extent of precipitation responses, particularly over the monsoon precipitation region. E1 (panel a) shows a pronounced positive precipitation response concentrated over the northern Indian subcontinent, with maximum increases exceeding 10 mm/day (dark red) in the Indo-Gangetic Plain. The wind field anomalies in E1 demonstrate enhanced westerly flows over the Arabian Sea and a strengthened cyclonic circulation over northern India. E2 (panel b), while showing some similarities in the broad-scale pattern, exhibits notable differences in both magnitude and spatial distribution of precipitation and circulation changes. While positive precipitation changes are still present over parts of northern India, they are much weaker and spatially less extensive. Moreover, E2 shows a more prominent negative precipitation anomaly over central India and the Bay of Bengal. The circulation anomalies in E2, though similarly westerly over the Arabian Sea, show a weaker cyclonic component over northern India and a different spatial organization of the wind field over the Indian peninsula. These differences, derived from different 10-member ensemble combinations, indicate that even with a moderate ensemble size of 10 members, simulations can produce opposed conclusions regarding dust impacts, though the contradictions are less severe compared to single-member simulations (Fig. 6).

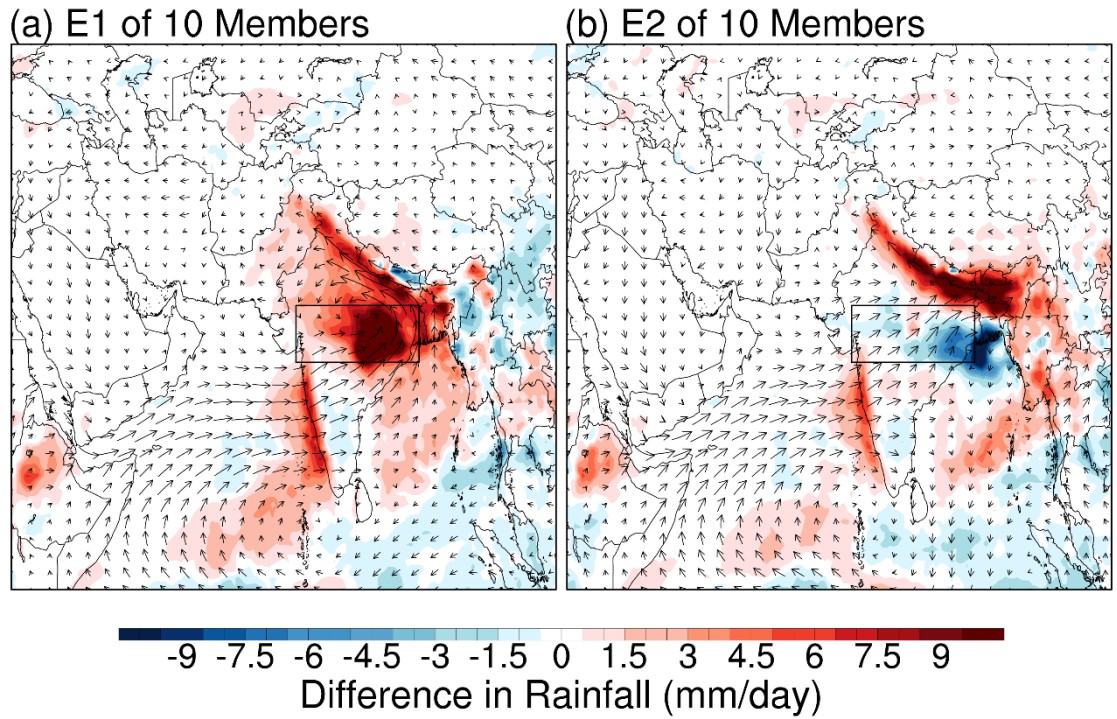

**Figure 8: The spatial distribution of dust-induced precipitation impacts for two extreme cases selected from 10,000 possible combinations of 10-member ensembles, representing the maximum (panel a) and minimum (panel b) area-averaged responses.**

Approach commonly employed with limited ensemble sizes involves statistical significance testing to validate simulation results. Figure 9 compares the spatial patterns of dust-induced precipitation changes from two 10-member ensemble combinations, with areas of statistical significance ($p < 0.05$) highlighted by purple stippling. The statistical significance of the differences is assessed using Student's t-test, performed at each grid cell by comparing 10 samples of ensemble member

values from the "Control" experiment against 10 corresponding samples from the "Sensitive" experiment, to determine if the results between the two experiments are significantly different. The left panel shows the average of the 10 members producing maximum area-averaged responses, featuring strong positive precipitation changes over the northern Indian subcontinent. The right panel, representing the average of the 10 members with minimum area-averaged responses, displays a notably different pattern with weaker positive changes over northern India and more extensive negative anomalies over central India and the

Bay of Bengal. Importantly, despite their contrasting precipitation patterns, both combinations show statistical significance (purple stippling) in key regions, even over regions with opposite dust-induced impacts on precipitation (such as marked by the black box). To determine whether these contradictory results of precipitation are caused by dust radiative forcings, we also calculate the corresponding dust TOA forcing difference of E1 and E2. The results show that, consistent with the high spatial coherence in dust AOD across ensemble members (Fig. 4), the dust-induced TOA radiative forcing differences between

contrasting subsets (e.g., E1 and E2) were found to be very small (Fig. S10). This analysis demonstrates that achieving statistical significance alone may not guarantee reliable representation of dust impacts when using small ensembles (e.g., only 10 members). Crucially, in practice, the specific subset of 10 members run in a study is essentially a random draw from the larger possible set. It could be any subset, including ones like E1 or E2 that produce statistically significant yet contradictory results. Rather than suggesting statistical tests are not meaningful, our results emphasize the importance of adequate ensemble

size to ensure robust characterization of aerosol impacts. This analysis is particularly relevant because 10-member ensembles (or less members) are widely used in current climate modeling studies due to computational resource limitations. However, as demonstrated by these discussions, such commonly used ensemble sizes may still be insufficient for robust characterization of dust-monsoon interactions. While 10-member ensembles represent a typical compromise between computational feasibility and scientific reliability in many studies, our results suggest that larger ensemble sizes might be necessary for more accurate

representation of dust-induced impacts on the monsoon system. It is crucial to emphasize that the ensemble size requirements discussed here are specific to the analysis of synoptic-scale processes within this 20-day simulation during the monsoon onset period. Studies focusing on longer-term climatological means (e.g., seasonal averages or multi-year averages) inherently integrate over more weather events. This temporal smoothing might accelerate the convergence towards a robust physical effect in function of ensemble size, which is a promising hypothesis that warrants systematic investigation in future studies.

Our findings on the necessity of larger ensembles therefore primarily apply to dust aerosol impacts on synoptic events, where the stochastic component of variability remains dominant and unresolved by temporal averaging.

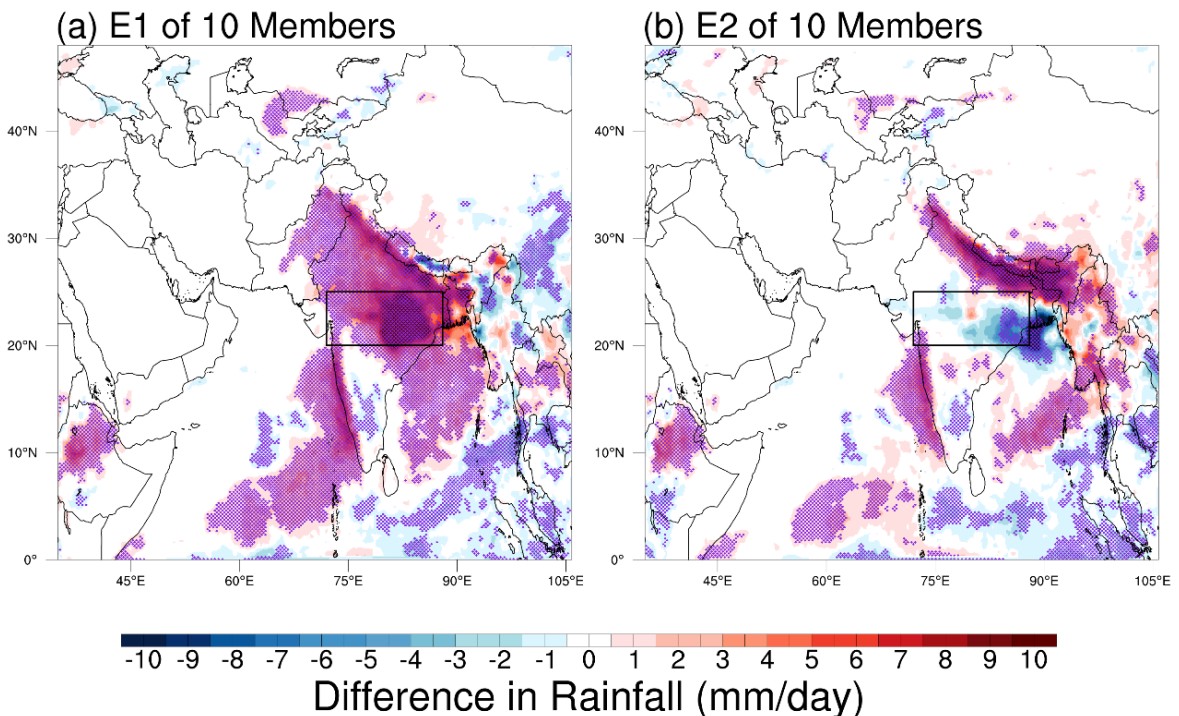

**Figure 9: 95% significance test (Student's t-test) results for two extreme cases selected from 10,000 possible combinations of 10-member ensembles, representing the maximum (panel a) and minimum (panel b) area-averaged responses.**

Figure 10 extends this analysis by examining the relationship between ensemble size and the range of simulated dust impacts. The panels are arranged to show the maximum (top row) and minimum (bottom row) area-averaged responses for increasing ensemble sizes from 1 to 40 members (1, 5, 10, 20, 30, and 40 are presented). For the monsoon depression region, ensembles with fewer than 30 members can produce substantially different, or even opposing, dust-induced impacts on precipitation and circulation patterns. By 30 members, the spatial patterns become notably more similar between maximum and minimum cases, with main differences reduced to the magnitude rather than the sign of the response. Notably, this sensitivity to initial conditions varies considerably by region. For instance, precipitation responses along India's western coast and the southern slopes of the Himalayas achieve reasonable convergence with as few as 5 ensemble members. These findings lead to several key conclusions: (1) For mesoscale weather systems, such as monsoon depressions, large ensemble sizes (approximately 30

members) are necessary to obtain robust simulations of dust aerosols effects. (2) For larger-scale processes, such as general monsoon circulation and moisture transport, smaller ensemble sizes (approximately 5 members) may suffice for accurate representation of dust impacts. (3) The chaotic nature of mesoscale systems likely depends on their dominant formation mechanisms—orographically-forced systems (the southern slopes of the Himalayas, for example) might be less sensitive to the initial conditions, which may show greater deterministic behavior. This scale-dependent and process-dependent requirement for ensemble size reflects the inherent predictability differences between synoptic-scale and mesoscale atmospheric processes in dust impacts studies. Please note that our findings of 30 members for mesoscale systems and 5 members for larger-scale processes are specific to our case study of dust effects on the ISM during June 10-30, 2016, and may vary for different aerosol types, regions, or seasons. The optimal ensemble size ultimately depends on the specific research questions, phenomena of interest.

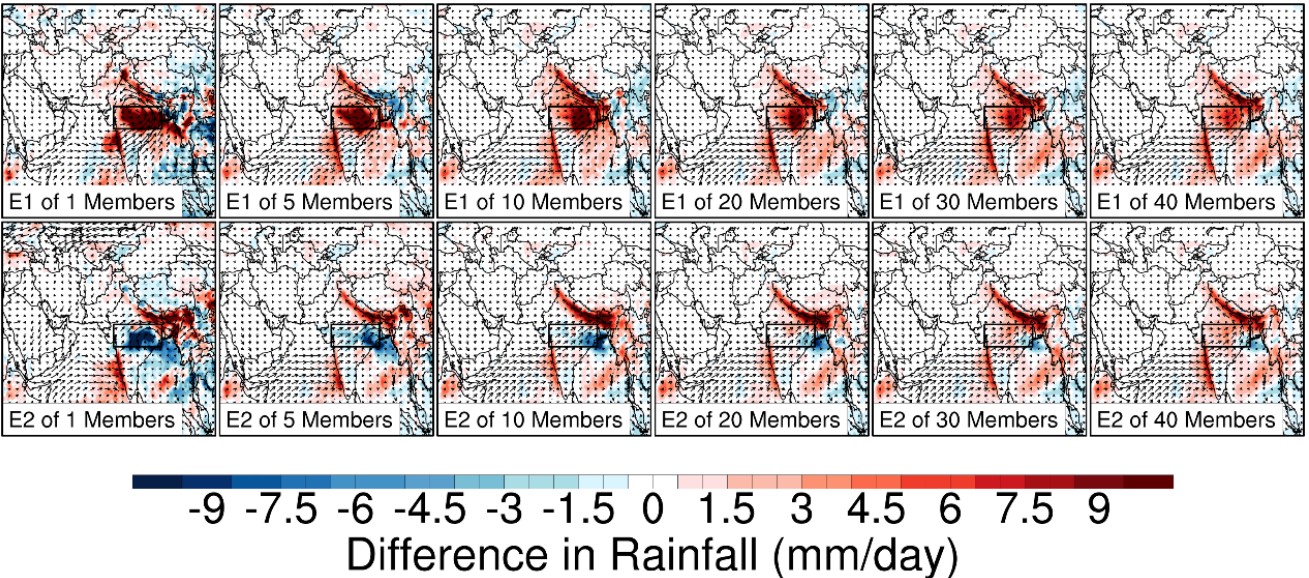

**Figure 10: The spatial distribution of dust-induced precipitation impacts for two extreme cases selected from possible combinations of 1, 5, 10, 20, 30, and 40-member ensembles, representing the maximum (E1 in top panels) and minimum (E2 in bottom panels) area-averaged responses.**

We extend a similar analysis to dust impacts on 850 hPa temperature following Figure 10. The results (as shown in Figure S11) indicate that temperature responses to dust aerosol forcing may converge with smaller ensemble sizes compared to precipitation responses. While we did not focus on temperature in this study, the observed patterns suggest that temperature fields could be used to isolate aerosol radiative effects with relatively modest ensemble sizes. This likely reflects that temperature responses more directly reflect the radiative perturbation from dust, whereas precipitation involves additional complex processes such as cloud microphysics, convective dynamics, and boundary layer interactions, which amplify the

influence of chaotic variability. The role of chaotic effects in modulating dust aerosol impacts across different climate variables and processes represents a compelling avenue for future research.

## 4 Summary and Discussions

This study investigates the role of chaotic effects in modulating complex interactions between dust aerosols and climate systems. We employed the iAMAS model to conduct large ensemble simulations with 50 members, aiming to bridge the gap in understanding uncertainties of simulating aerosols impacts introduced by the chaotic effect, and to distinguish between physical and chaotic effects in simulating aerosol impacts. Our results demonstrate that dust emissions from Central-East Asia significantly influence the ISM monsoon, with pronounced effects on monsoon circulation and precipitation patterns. The results reveal that dust aerosols strengthen the southwesterly monsoon flow from the Arabian Sea toward the Indian subcontinent, leading to enhanced precipitation along the western Indian coast and the western Himalayan foothills, which is consistent among most ensemble members and aligns with previous studies (Vinoj et al., 2014; Jin et al., 2015; Lau et al., 2017).

However, the results also reveal the critical role of chaotic effects in dust-monsoon interactions. The simulated dust impacts on regional systems, such as monsoon depressions, exhibit significant uncertainties induced by chaotic effects. Even with 10-member ensembles, a commonly used ensemble size in many studies, simulations can produce fundamentally different or even opposing conclusions about dust impacts on regional rainfall patterns. Statistical significance testing also be proved insufficient for establishing result robustness, as demonstrated by cases where contradictory results both achieve statistical significance.

Moreover, our analysis reveals that the magnitude of chaotic effects diminishes with increasing ensemble size, which is proportional to $N^{-\frac{1}{2}}$, suggesting that larger ensembles are necessary for physical characterization of dust-monsoon interactions at regional scales. The required ensemble size exhibits strong spatial dependence, reflecting different predictability characteristics of atmospheric processes at various scales. While large-scale features like monsoon circulation can be reliably simulated with relatively few ensemble members (e.g., 5 members), mesoscale features such as monsoon depressions require substantially larger ensembles (e.g., 30 members or larger) to achieve convergence. This scale-dependent behavior suggests that studies focusing on regional-scale processes may need to carefully consider ensemble size requirements based on their specific phenomena of interest.

While this investigation focuses on dust-monsoon interactions over ISM, the implications extend beyond this specific case study. Our findings suggest that chaotic effects should be carefully considered in broader aerosol-climate interaction studies, particularly those focusing on regional and mesoscale processes. The contrasting results reported in previous studies about aerosol-weather interactions at regional scale might reflect insufficient ensemble sizes rather than fundamental disagreements

in physical mechanisms. This raises important questions about the robustness of conclusions drawn from existing aerosol-climate studies that rely on limited ensemble sizes or single simulations.

Several limitations and future research directions emerge from this work. First, our analysis excluded potentially important processes like dust-induced ice nucleation that could further amplify chaotic effects through cloud-precipitation feedbacks. Second, the required ensemble size likely varies across different geographical regions and meteorological systems - areas with

530 strong mesoscale processes or complex topography might require larger ensembles than regions dominated by large-scale circulation patterns. Third, different aerosol species with distinct radiative and microphysical properties may exhibit varying sensitivities to chaotic effects. Fourth, longer model integrations might dampen the chaotic effects seen at short time scale by temporal averaging, which deserves further research to be carefully addressed. As we only conducted experiments for 20 days, the ensemble size suggested in this study might bigger than that needed for multi-year or seasonal mean

studies. Finally, it is important to consider the implications of prescribing SST. The absence of interactive ocean feedbacks (e.g., damping of atmospheric fluctuations through SST changes) may influence the development of internal variability (the constant supply of moisture and energy from the ocean surface could potentially enhance the growth and stochasticity of perturbations compared to a coupled system where the ocean would respond and potentially dampen atmospheric fluctuations). This suggests that the chaotic effects might be larger in our experimental setup than they would be in a fully coupled system

with interactive ocean feedbacks. Consequently, the ensemble size requirements we derived for robustly detecting dust aerosol impacts could be viewed as conservative estimates in the context of coupled modeling. Future work should explore these chaotic effects across different regions, aerosol types, and meteorological systems to develop more comprehensive guidelines for chaotic effects on aerosol impacts studies. This is particularly important for regions with strong aerosol-weather interactions with complex mechanisms. Furthermore, our team plans to extend this analysis to seasonal and multi-year timescales to

quantify how temporal averaging affects the required ensemble size and to characterize the chaotic effects across different timescales. Additionally, developing more efficient methods to account for chaotic effects while maintaining computational feasibility remains an important challenge, especially for global climate simulations where large ensembles may be computationally prohibitive.

**Code availability**

The used version of the iAMAS model is openly available at Zenodo (https://doi.org/10.5281/zenodo.7571299).

**Data availability**

ERA5 data are available from the Copernicus Climate Data Store (https://cds.climate.copernicus.eu). The MISR Level 3 aerosol products are available from the NASA Atmospheric Science Data Center (https://asdc.larc.nasa.gov). CMORPH V1.0 high-resolution global precipitation data can be accessed through NCAR's Research Data Archive
(https://rda.ucar.edu).

**Author contribution**

JF, SL, and CZ designed the experiments. JF and CZ conducted the experiments. JF performed the analysis and wrote the manuscript draft. CZ, JF, JG, DL, MX, and JF (Jie Feng) reviewed and edited the manuscript.

**Competing interests**

The authors declare that they have no conflict of interest.

**Acknowledgements**

This research was supported by the National Key Research and Development Program of China (2024YFF0811200, 2022YFC3700701), the Strategic Priority Research Program of Chinese Academy of Sciences (XDB0500303), the USTC Research Funds of the Double First-Class Initiative (YD2080002007, KY2080000114), the Natural Science Foundation of
Anhui (2208085UQ09, 2208085UQ02), the Science and Technology Innovation Project of Laoshan Laboratory (LSKJ202300305) and the Postdoctoral Fellowship Program of CPSF(GZC20250188). This study used computing resources from the Supercomputing Center of the University of Science and Technology of China (USTC), the National Key Scientific and Technological Infrastructure project "Earth System Numerical Simulation Facility" (EarthLab), and the Qingdao Supercomputing and Big Data Center.

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
