# Peer review of "Dust impacts on Indian summer monsoon: chaotic or physical effect?"

_EGUsphere, 2024_

## Author Comment (AC1)

**Response to Referee #2**

*This manuscript investigates the chaotic vs. physical effects of dust aerosols on Indian Summer Monsoon (ISM) precipitation using large ensemble simulations with the iAMAS model. The study focuses on a 20-day period in June 2016 and quantifies the spread and convergence of dust-induced impacts using 50-member ensembles. The authors use the Indian Summer Monsoon (ISM) system as a case study to show that even with the same physical forcing (e.g., dust aerosol), the simulated response varies widely due to initial condition perturbations. The novel aspect lies in highlighting the limitations of small ensemble sizes in drawing robust conclusions about aerosol effects.*

**Response:** We sincerely thank Referee #2 for the careful review and insightful summary of our manuscript. We have carefully addressed all your specific comments and suggestions in the revised manuscript, as detailed in our point-by-point responses below. We believe these revisions have helped us improve the clarity of our presentation and strengthen our conclusions and hope that they adequately address your concerns.

**The major comments are:**

- *1. The final paragraph of the introduction should more clearly articulate the main objectives of the study and provide a concise roadmap of the manuscript's structure. Currently, the paragraph combines motivation and definitions without explicitly stating specific research questions or outlining the paper's structure.*

**Response:** We sincerely thank the reviewer for this suggestion to enhance the clarity of our introduction's final paragraph. We have revised this section and added our main objectives and provide a clear roadmap as follows: "This study has three primary objectives: (1) to quantify the uncertainties in simulating aerosol impacts introduced by chaotic effects, (2) to distinguish between physical and chaotic effects in the dust aerosol impacts on ISM system, and (3) to determine whether simulated aerosol impacts on the ISM are predominantly driven by physical processes or significantly influenced by chaotic behaviors. We define the "physical effect" as the deterministic response of meteorological fields to aerosols that remains consistent across ensemble members despite initial condition perturbations. The ensemble-mean approximates this underlying physical effect by averaging out chaotic influences. Conversely, the "chaotic effect" represents internally generated variations arising from initial condition perturbations, manifested as the spread among ensemble members (Feng et al., 2024a).

The remainder of this paper is structured as follows: Section 2 describes our methodology, including the iAMAS model employed (Section 2.1), experiments configurations and methods for generating perturbed initial conditions (Section 2.2), and observational datasets used for validation (Section 2.3). Section 3 presents our analysis of chaotic effects on dust aerosol impacts on the ISM and discusses the relationship between ensemble size and chaotic uncertainties. Section 4 provides conclusions and summarizes the implications of our findings and discusses the limitations of this study."

- ***2. The description of the iAMAS model is currently scattered within the introduction, primarily through citations to previous studies. However, a dedicated and concise model description paragraph is missing from the Methodology section, which is where readers expect to find details about the modeling framework used in the experiments. I recommend moving the relevant model description content from the introduction to Section 2.1, ensuring it covers key features (e.g., dynamics, resolution, aerosol treatment, radiation, and physics schemes) in a self-contained manner. This will improve the clarity and reproducibility of the study.***

**Response:** We sincerely appreciate the reviewer's suggestion regarding the organization of the iAMAS model description. We have added more descriptions of iAMAS's critical components in the revised Section 2.1 to improve the clarity and reproducibility of our study as follows:

[revised manuscript text omitted]

- **3. The manuscript lacks clarity on aerosol treatment, especially dust. Please specify:**
  - ✓ *Whether both direct (radiative) and indirect (cloud) effects are included. If only direct effects (ARI) are used, this should be clearly stated and justified.*
  - ✓ *Whether aerosol-cloud interactions (ACI) are active, and if not, why.*
  - ✓ *Whether aerosols are internally or externally mixed, and what assumptions are made regarding their optical and hygroscopic properties.*

  *I recommend that the authors include a dedicated subsection or an expanded paragraph in Section 2 covering these aerosol processes in sufficient detail.*

**Response:** We sincerely appreciate the reviewer's insightful suggestions regarding aerosol process specification. As addressed in our prior response, we have expanded Section 2.1 to detailed document our treatment of aerosols. To directly answer the reviewer's queries: 1. The ACI and ARI are both used in this study. 2. ACI is active but the IN feedbacks are deactivated because these calculations in iAMAS are not fully evaluated at this stage. 3. Aerosols are internally mixed and the optical and hygroscopic properties are also described in the revised text. Please refer to the Methodology of revised text for details.

- **4. The choice to simulate only 20 days during the early monsoon season (June 10–30, 2016) warrants further justification. This short time frame captures only the monsoon onset and not the full seasonal evolution, intraseasonal variability, or withdrawal phase. While the period may have been chosen to isolate certain synoptic features or reduce computational cost, the manuscript should explicitly state the scientific rationale for selecting this specific window. Additionally, it would strengthen the study to discuss how representative this period is of broader monsoon-dust interactions. If this is intended as a case study, that should be clearly stated to avoid overgeneralization of the results.**

**Response:** Thanks a lot for this critical point. We have clarified in the revised Section 2 text as: "The simulations covered the period from June 10 to June 30, 2016, focusing on a specific intense rainfall period occurring during the 2016 Indian summer monsoon season. To be clarified, this period does not cover the entire dust-ISM interactions throughout the monsoon season or across different years. We selected this specific period as it features a monsoon onset period with monsoon depression system that is particularly sensitive to aerosol impacts, making it suitable for investigating physical and chaotic effects. This approach also balances computational costs (necessitated by the large number of ensemble experiments) with scientific objectives, though we recognize that longer-term simulations would be valuable for future work to capture the full range of dust-ISM interaction."

Besides, we have added additional discussions in the revised Introduction section as: "While substantial progress has been made in characterizing dust-monsoon interactions, most previous studies have focused on the mature monsoon season (July-August), during which atmospheric circulation is more stable and convective systems are already well established. In contrast, the onset phase is dynamically transitional and thus more sensitive to radiative and thermodynamic perturbations. During this transition, atmospheric circulation is dynamically unstable, the Intertropical Convergence Zone (ITCZ) and low-level jets are reorganizing, and synoptic systems such as monsoon depressions are forming. Under such complex conditions, dust-induced heating may exert outsized influence. Furthermore, to investigate the influence of chaotic effects of dust impacts, we plan to conduct a large ensemble of experiments with 50 members, which demands substantial computational resources. Given that dust may exert a pronounced influence during the onset period and to manage the computational resource constraints, we select only the onset period of the ISM in 2016 (June 10–30) as our simulation period."

And we have avoided overgeneralization of the results by adding: "To be clarified, our results on precipitation response patterns reflect this specific meteorological situation (Jun 10 to Jun 30, 2016), and the large effect we document here specifically applies to dust's role during the monsoon onset period in modulating the formation of monsoon depression systems during favorable meteorological conditions, rather than representing a general dust-monsoon interaction magnitude that could be extrapolated to seasonal or climatological time scales." in the revised text in Section 3 (Line 350) and "It is crucial to emphasize that the ensemble size requirements discussed here are specific to the analysis of synoptic-scale processes within this 20-day simulation during the monsoon onset period. Studies focusing on longer-term climatological means (e.g., seasonal averages or multi-year averages) inherently integrate over more weather events. This temporal smoothing might accelerate the convergence towards a robust physical effect in function of ensemble size, which is a promising

hypothesis that warrants systematic investigation in future studies. Our findings on the necessity of larger ensembles therefore primarily apply to dust aerosol impacts on synoptic events, where the stochastic component of variability remains dominant and unresolved by temporal averaging." in Section 3 (Line 457).

- *5. Figure 3c (AOD from the "Sensitive" simulation) appears to show nearly no aerosol loading over much of South Asia, including the Indo-Gangetic Plain — a region known for high anthropogenic aerosol concentrations even during the monsoon period. Since the "Sensitive" case only excludes Arabian dust emissions, anthropogenic aerosol emissions should still be present in the simulation or was it only dust emissions enabled?*

**Response:** Sorry for the confusion. Actually, only dust emissions are enabled in this study. We have clarified this point in the revised manuscript's methodology section as: "In the experiments conducted for this study, only dust aerosols are included to isolate their effects from those of other aerosols."

- *6. Figure 10 suggests that ensemble sizes beyond 30 members yield only marginal improvements in the convergence of dust-induced precipitation responses. Given the computational cost associated with running large ensembles, could the authors clarify whether they consider 30 members as an optimal threshold for similar studies? Additionally, do they recommend any specific criteria or diagnostics to determine when further increases in ensemble size (e.g., to 40 or more) are justified?*

**Response:** Thank you for this insightful question regarding optimal ensemble sizes. We would like to clarify that the threshold of 30 members suggested by Figure 10 should not be interpreted as a universally optimal value for all aerosol impacts studies. Rather, our analysis demonstrates a scale-dependent relationship between required ensemble size and the meteorological phenomena being studied.

Our results indicate that different atmospheric processes require different minimum ensemble sizes to achieve robust results:

For mesoscale weather systems like the monsoon depression examined in our case study, we found that ensembles with fewer than 30 members could produce substantially different or even opposing dust-induced impacts. With approximately 30 members, the spatial patterns of responses showed much better convergence, with differences mainly in magnitude rather than sign.

In contrast, for larger-scale processes such as general monsoon circulation and precipitation along India's western coast and the southern slopes of the Himalayas, reasonable convergence was achieved with as few as 5 ensemble members.

This scale-dependence reflects the inherent predictability differences between large-scale and mesoscale atmospheric processes. Smaller-scale phenomena generally exhibit greater sensitivity to initial conditions and thus require larger ensembles to robustly characterize their responses to aerosol forcing.

We also explicitly state in the revised manuscript as: "Please note that our findings of 30 members for mesoscale systems and 5 members for larger-scale processes are specific to our case study of dust effects on the ISM during June 10-30, 2016, and may vary for different aerosol types, regions, or seasons. The optimal ensemble size ultimately depends on the specific research questions, phenomena of interest."

- *7. Would the authors expect similar sensitivity and ensemble size requirements if the primary response variable were temperature rather than precipitation? Could temperature fields, given their typically lower chaotic variance, be used to isolate aerosol radiative effects with smaller ensemble sizes?*

**Response:** Thank you for this insightful question about temperature fields as response variables. You raise an important point that highlights the variable sensitivity of different meteorological parameters to chaotic effects.

As shown in Figure R2 below, temperature fields indeed exhibit considerably less chaotic variance compared to precipitation fields. This aligns with fundamental atmospheric dynamics - temperature fields tend to be more spatially coherent and temporally stable than precipitation.

[Figure]

**Figure R2.** The spatial distribution of dust-induced temperature at 850hPa impacts for two extreme cases selected from possible combinations of 1, 5, 10, 20, 30, and 40-member ensembles, representing the maximum (E1 in top panels) and minimum (E2 in bottom panels) area-averaged responses.

Given these results, we would expect temperature responses to dust aerosol forcing to converge with smaller ensemble sizes compared to precipitation responses. While we did not explicitly test convergence thresholds for temperature in this study, our results suggest that temperature fields could likely be used to isolate aerosol radiative effects with smaller ensemble sizes - perhaps around 10 members or even fewer for robust characterization of the temperature response.

This supports your suggestion that temperature fields could be a more computationally efficient way to isolate certain aerosol radiative effects. Temperature responses directly reflect the radiative perturbation from dust, whereas precipitation responses involve additional complex processes including cloud microphysics, convective dynamics, and boundary layer interactions, all of which amplify the influence of chaotic variability.

In our current study, we tend to focused on precipitation as one of the most challenging variables to characterize robustly, providing an estimate of required ensemble sizes. For comprehensive aerosol impact studies targeting multiple variables and processes, ensemble size requirements would likely be dictated by the most chaotically sensitive variables of interest.

This differential sensitivity across variables is an interesting avenue for future work, and we appreciate your suggestion to consider how ensemble requirements might be optimized depending on the specific response variables being investigated.

We have added some discussions in the revised manuscript as: "We extend a similar analysis to dust impacts on 850 hPa temperature following Figure 10. The results (as shown in Figure S11) indicate that temperature responses to dust aerosol forcing may converge with smaller ensemble sizes compared to precipitation responses. While we did not focus on temperature in this study, the observed patterns suggest that temperature fields could be used to isolate aerosol radiative effects with relatively modest ensemble sizes. This likely reflects that temperature responses more directly reflect the radiative perturbation from dust, whereas precipitation involves additional complex processes such as cloud microphysics, convective dynamics, and boundary layer interactions, which amplify the influence of chaotic variability. The role of chaotic effects in modulating dust aerosol impacts across different climate variables and processes represents a compelling avenue for future research."

- *8. Since the study focuses predominantly on dust aerosols and specifically targets the Indian Summer Monsoon, I recommend the review title to more accurately reflect this focus.*

**Response:** Thanks a lot for reviewer's insightful recommendation to emphasize the specific role of dust aerosols in the Indian Summer Monsoon. The new title now directly reflects this core focus: "Dust impacts on Indian summer monsoon: chaotic or physical effect?".

---

## Author Comment (AC2)

**Response to Referee #1**

*This study proposes an analysis of the role of weather and climate stochasticity impacting the response of the Indian monsoon precipitation to Arabian dust regional radiative forcing. It relies on the statistical analysis of a large ensemble of regional short-term simulations based on a global, dust interactive, atmospheric model to discuss the regional significance of 'true' physical dust induced response vs purely chaotic internal variability response. It also examines the number of ensemble members needed to achieve converging and robust results. It outlines that some sub-region like central India where precipitation response depends on meso-scale weather system organization are more prone to internal variability compared to other region where the impact of large scale flow dominates. The study concludes that most of existing studies looking at dust impact on Indian monsoon precipitation did not properly account for these effects, explaining divergence in results especially for central India.*

*The topic is definitely relevant to ACP, the paper and methodology are in general appropriate, clearly written. Although the topic of stochastic effects / internal variability affecting sensitivity and climate change studies has been explored, seeing it applied on the specific issue of dust /Indian monsoon interaction is definitely interesting in my opinion. Overall I find the paper suitable for publication, after taking into account the following points :*

**Response:** We sincerely thank Referee #1 for the thorough and constructive review of our manuscript. We appreciate your positive assessment of our work's relevance to ACP and the recognition that applying stochastic/internal variability analysis to the specific issue of dust-Indian monsoon interaction provides valuable insights. We have carefully considered all your comments and suggestions, as detailed in our point-by-point responses below. We believe these revisions have strengthened our manuscript and hope that they adequately address your concerns.

**Major comment :**

- *My main criticism concerns the Authors generalizing their conclusion a bit quickly regarding other existing climatic studies. Indeed the proposed simulation protocol uses a lot of members (50), but it also explore the dust induced response on a relatively short time scale of 20 days, characteristic of June of a given year. So each members includes a limited number of meso-scale events for example. Climatic studies are often based on multi-year simulation and examine the impacts of dust on seasonal and yearly averaged precipitation, they thus includes many events and part of the internal variability effect might be smoothed out when averaging. I am not saying that the internal variability does not affect these studies, I am sure it does, but perhaps the convergence toward a consistent physical signal is achieved faster (i.e. with less members) when dealing with climate length simulations. I advise the Authors to be cautious when making conclusion "at climate scale" and regarding "the Indian Monsoon", or to clearly demonstrate how their results can be be generalized.*

**Response:** We greatly appreciate the reviewer's insightful comment regarding the generalization of our conclusions to longer climatological studies. We fully agree that our 20-day simulation protocol differs fundamentally from multi-year climatological studies in terms of temporal averaging and the number of weather events sampled.

In response to this concern, we have revised our manuscript to be more cautious when making conclusions about "climate scale" effects and "the Indian Monsoon" in general. Specifically, we have:

Added clarification in the methodology section (Section 2.2.1) that our results apply specifically to synoptic-scale processes within a 20-day window as: "The simulations covered the period from June 10 to June 30, 2016, focusing on a specific intense rainfall period occurring during the 2016 Indian summer monsoon season. To be clarified, this period does not cover the entire dust-ISM interactions throughout the monsoon season or across different years. We selected this specific period as it features a monsoon onset period with monsoon depression system that is particularly sensitive to aerosol impacts, making it suitable for investigating physical and chaotic effects. This approach also balances computational costs (necessitated by the large number of ensemble experiments) with scientific objectives, though we recognize that longer-term simulations would be valuable for future work to capture the full range of dust-monsoon interaction."

Revised our discussion in Section 3 to explicitly acknowledge that: "To be clarified, our results on precipitation response patterns reflect this specific meteorological situation (Jun 10 to Jun 30, 2016), and the large effect we document here specifically applies to dust's role during the monsoon onset period in modulating the formation of monsoon depression systems during favorable meteorological conditions, rather than representing a general dust-monsoon interaction magnitude that could be extrapolated to seasonal or climatological time scales." and "It is crucial to emphasize that the ensemble size requirements discussed here are specific to the analysis of synoptic-scale processes within this 20-day simulation during the monsoon onset period. Studies focusing on longer-term climatological means (e.g., seasonal averages or multi-year averages) inherently integrate over more weather events. This temporal smoothing might accelerate the convergence towards a robust physical effect in function of ensemble size, which is a promising hypothesis that warrants systematic investigation in future studies. Our findings on the necessity of larger ensembles therefore primarily apply to dust aerosol impacts on synoptic events, where the stochastic component of variability remains dominant and unresolved by temporal averaging."

Modified our conclusions in Section 4 to emphasize that our findings on ensemble size requirements specifically apply to resolving dust impacts on synoptic-scale monsoon features, might not necessarily to climatological studies where temporal averaging provides additional robustness as: "Fourth, longer model integrations might dampen the chaotic effects seen at short time scale by temporal averaging, which deserves further research to be carefully addressed. As we only conducted experiments for 20 days, the ensemble size suggested in this study might bigger than that needed for multi-year or seasonal mean studies." and emphasizes the need for future research at longer timescales in Sections 4 as: "Furthermore, our team plans to extend this analysis to seasonal and multi-year timescales to quantify how temporal averaging affects the required ensemble size and to characterize the chaotic effects across different timescales."

We believe these revisions achieved to contextualize our findings while maintaining the value of our insights for understanding short-term, event-scale processes.

- ***Regarding the radiative forcing and significant precipitation response obtained through a robust ensemble average, it would be also good to recall that the corresponding patterns and magnitude reflect a specific June 10-30 2016 situation, which does not cover the entire variability of dust - Indian monsoon interactions.***

**Response:** We appreciate the reviewer's valuable comment on the temporal scope of our study. We fully agree that our findings reflect a specific period (June 10-30, 2016) and do not capture the entire variability of dust-Indian monsoon interactions. In response, we have revised our manuscript to explicitly acknowledge this limitation with the following text: "The simulations covered the period from June 10 to June 30, 2016, focusing on a specific intense rainfall period occurring during the 2016 Indian summer monsoon season. To be clarified, this period does not cover the entire dust-ISM interactions throughout the monsoon season or across different years. We selected this specific period as it features a monsoon onset period with monsoon depression system that is particularly sensitive to aerosol impacts, making it suitable for investigating physical and chaotic effects. This approach also balances computational costs (necessitated by the large number of ensemble experiments) with scientific objectives, though we recognize that longer-term simulations would be valuable for future work to capture the full range of dust-ISM interaction."

We acknowledge that our results on radiative forcing and precipitation response patterns reflect this specific meteorological situation, and the magnitude and spatial patterns of dust-monsoon interactions may vary during other phases of the monsoon or in different years. And this is also clarified in the revised text in Section 3 (Line 350): "To be clarified, our results on precipitation response patterns reflect this specific meteorological situation (Jun 10 to Jun 30, 2016), and the large effect we document here specifically applies to dust's role during the monsoon onset period in modulating the formation of monsoon depression systems during favorable meteorological conditions, rather than representing a general dust-monsoon interaction magnitude that could be extrapolated to seasonal or climatological time scales."

- ***Finally I did not really understand the method for discussing the validity of statistical significance tests in few ensemble member studies, please see specific comments.***

**Response:** The method is Student's t-test. For more details, please refer to our response to the specific comments below.

**Minor/specific comments :**

- ***Title : Perhaps a title more focused on dust and the region of study would be more appropriate.***

**Response:** We sincerely thank Reviewer 1 for the constructive suggestion to refine the title for greater focus. We have revised the title to explicitly highlight dust aerosols and the Indian Summer Monsoon as: "Dust impacts on Indian summer monsoon: chaotic or physical effect?"

- ***L45 -50 ; An other useful reference focusing on regional climate models Internal variability. O'Brien et al.. Clim Dyn 37, 1111–1118 (2011). https://doi.org/10.1007/s00382-010-0900-5***

**Response:** We appreciate the reviewer's valuable suggestion regarding the inclusion of O'Brien et al.'s work. This study provides important insights into understanding internal variability in regional climate simulations, which is highly relevant to our research focus. We have incorporated it into the revised text as: "O'Brien et al., (2011) indicated that intrinsic variability (IV) of precipitation in

regional climate models can be large enough to violate the assumptions of sensitivity study." in Section 1.

- *L141 : the simulations are 20 days long, representing a specific month of a specific year. Can we say this protocol « captures the Indian monsoon » ? To me the experiment is closer to a meteorological experiment than a climatological experiment. The intra-seasonal and interannual variability of the ISM are here not captured.*

**Response:** We sincerely thank the reviewer for raising this point regarding the temporal scope of our simulations and the interpretation of capturing the Indian summer monsoon (ISM). We acknowledge that the phrasing in the original manuscript (L141: "capturing the Indian summer monsoon season") was potentially misleading in implying a comprehensive representation of the ISM. We have revised the text to make it more accurate as: "The simulations covered the period from June 10 to June 30, 2016, focusing on a specific intense rainfall period occurring during the 2016 Indian summer monsoon season. To be clarified, this period does not cover the entire dust-ISM interactions throughout the monsoon season or across different years. We selected this specific period as it features a monsoon onset period with monsoon depression system that is particularly sensitive to aerosol impacts, making it suitable for investigating physical and chaotic effects. This approach also balances computational costs (necessitated by the large number of ensemble experiments) with scientific objectives, though we recognize that longer-term simulations would be valuable for future work to capture the full range of dust-ISM interaction."

Besides, we have added additional discussions in the revised Introduction section as: "While substantial progress has been made in characterizing dust-monsoon interactions, most previous studies have focused on the mature monsoon season (July-August), during which atmospheric circulation is more stable and convective systems are already well established. In contrast, the onset phase is dynamically transitional and thus more sensitive to radiative and thermodynamic perturbations. During this transition, atmospheric circulation is dynamically unstable, the Intertropical Convergence Zone (ITCZ) and low-level jets are reorganizing, and synoptic systems such as monsoon depressions are forming. Under such complex conditions, dust-induced heating may exert outsized influence. Furthermore, to investigate the influence of chaotic effects of dust impacts, we plan to conduct a large ensemble of experiments with 50 members, which demands substantial computational resources. Given that dust may exert a pronounced influence during the onset period and to manage the computational resource constraints, we select only the onset period of the ISM in 2016 (June 10–30) as our simulation period."

- *L150 : Could you please mention at this stage if only dust radiative effects or both dust radiative and microphysical effects are taken into account in the experiments ? I understand it was stated in the conclusion.*

**Response:** Thanks for the suggestion. We realized that we missed some critical descriptions in the method section. Both dust radiative and microphysical effects are taken into account in the experiments. We have added some detailed descriptions of the model we used and the experiments in the revised Section 2.1 as: "Aerosol-cloud interaction (ACI) is implemented in the model based on the method described by (Gustafson et al., 2007) for calculating the activation and resuspension between dry aerosols and cloud droplets. Aerosol activation (or droplet nucleation) is based on a maximum supersaturation determined from a Gaussian spectrum of updraft velocities, similar to the

methodology used in (Ghan et al., 2001). The activated droplet number is then coupled with the Thompson microphysics scheme. In this way, aerosols can affect cloud droplet number, and clouds can also alter aerosol concentration through aqueous processes and wet scavenging. The hygroscopicity of dust aerosols are assumed to be 0.10 in this study. Within the Thompson cloud microphysics scheme, the number of ice nucleation (IN) in mixing-phase clouds from dust is calculated following the formula proposed by DeMott et al.(DeMott et al., 2010). This study only considers the wet scavenging process of activated dust aerosols into cloud droplet, ignoring the conversion of dust into IN because the IN feedback calculations are not fully evaluated in iAMAS at this stage. iAMAS also incorporates the aerosol-radiation interaction (ARI). Following the new method proposed by Feng et al., (2025), aerosol optical properties are computed and coupled with the RRTMG radiation scheme for both shortwave and longwave bands. For dust aerosols, this study utilizes the Optical Properties of Aerosols and Clouds (OPAC) dataset (Hess et al., 1998) to provide their shortwave and longwave refractive indices."

- *Question: Despite the simulation time scale being relatively short and since IV develops from small perturbations, can the fact that SST are forced in your experiments affect the noise to signal ratio (and so the relative impact of internal variability) ? Fixed SST creates basically a constant supply of energy and moisture for the perturbations to develop without consistent dampening, perhaps this is likely to enhance stochasticity especially in convective regions. This would also be a contextual difference with climatic studies which consider an interactive ocean /SST.*

**Response:** We thank the reviewer for raising this critical question regarding the potential impact of prescribing sea surface temperatures (SST). We choose to use prescribed SST based on these reasons: 1. The simulations cover only 20 days. Over such a short period, the characteristic time scale for significant SST changes is typically longer (weeks to months). Prescribing observed daily SST is a standard and widely accepted approach; 2. We believe that the monsoon response to SST changes is very important (perhaps more important than the interaction we are documenting). However, our primary goal was to investigate the aerosols' effect on monsoon during this event. Fixing the SST boundary condition helps to isolate the aerosol effects by removing the complex feedbacks between the atmosphere and ocean. This allows us to more cleanly attribute the simulated variability to atmospheric mechanisms rather than coupled interactions. We have added further explanation in the revised text as: "Sea surface temperatures, prescribed from the ERA5 reanalysis dataset, were updated every 6 hours throughout the simulation period. This approach is common for short-term atmospheric process studies as the simulation period (20 days) is short compared to typical SST adjustment timescales. Besides, since SST is prescribed, the model differences will only be attributed to dust aerosol effects associated with aerosol-monsoon interaction."

Regarding the potential amplification of IV and noise-to-signal ratio, we agree that the absence of ocean damping feedbacks under fixed SST conditions could potentially allow small-scale atmospheric perturbations to grow more freely, especially in regions of strong convection. This might lead to an enhancement of the simulated stochasticity (the "noise" component of IV) compared to a coupled system where the ocean would respond and potentially dampen atmospheric fluctuations. Therefore, we have added these discussions in the revised Section 4 (Summary and Discussions) to clarify our work's limitations as: "Finally, it is important to consider the

implications of prescribing SST. The absence of interactive ocean feedbacks (e.g., damping of atmospheric fluctuations through SST changes) may influence the development of internal variability (the constant supply of moisture and energy from the ocean surface could potentially enhance the growth and stochasticity of perturbations compared to a coupled system where the ocean would respond and potentially dampen atmospheric fluctuations). This suggests that the chaotic effects might be larger in our experimental setup than they would be in a fully coupled system with interactive ocean feedbacks. Consequently, the ensemble size requirements we derived for robustly detecting dust aerosol impacts could be viewed as conservative estimates in the context of coupled modeling."

- *2.2.2 Generating Perturbed Initial Conditions for Ensembles: It seems that two distinct perturbation protocols are presented but I did not really understand why at this stage. Are they compared later on ?*

**Response:** We apologize for any confusion. To clarify, Section 2.2.2 describes a single protocol. The first part outlines the original Breeding of Growing Modes (BGM) method, while the second part details our adaptation of BGM for the iAMAS SCVTs grid, providing more details than the first part. We appreciate the reviewer's comment highlighting how this structure could be unclear to readers. In response, we have simplified the section by removing the steps description of the original BGM method; only the details of our adapted approach are now presented in the second paragraph. Please refer to the revised Section 2.2.2 in the updated manuscript.

- *L 295 : This robust effect of dust on precipitation (100% enhancement) is here quite large and significant. This is a strong result that would need to be more discussed in light of other studies. As I mentioned earlier, caution should be taken regarding how this result can be representative of "dust impact on monsoon" regarding the time-scale addressed.*

**Response:** Thank you for highlighting this important result and the need for broader context. We agree that the ~100% precipitation enhancement due to dust is indeed a substantial effect that warrants careful discussion. We have added more discussions in the revised text as: "The large magnitude of this dust-induced precipitation change can be attributed to the specific meteorological mechanism we investigated: dust aerosols' influence on monsoon depression formation during the monsoon onset. As we discussed in our analysis of individual ensemble members in Section 3.1, dust plays a critical role in determining whether monsoon depression-associated precipitation patterns develop successfully in our simulations. This binary-like behavior—where dust presence can influence whether or not a monsoon depression system forms—explains the large precipitation difference we observe. Monsoon depressions are known to produce rainfalls, capable of generating several mm/day of precipitation over extensive areas (Srivastava et al., 2017). Therefore, the difference between successfully simulating versus missing such a system naturally leads to substantial percentage changes in regional precipitation."

We acknowledge the reviewer's important caution about temporal representativeness. Our result reflects dust impacts during a specific 20-day period featuring conditions conducive to monsoon depression development, and we have added text emphasizing that this magnitude of dust impact may not be representative of dust-monsoon interactions across longer time scales or different synoptic conditions as: "To be clarified, our results on precipitation response patterns reflect this

specific meteorological situation (Jun 10 to Jun 30, 2016), and the large effect we document here specifically applies to dust's role during the monsoon onset period in modulating the formation of monsoon depression systems during favorable meteorological conditions, rather than representing a general dust-monsoon interaction magnitude that could be extrapolated to seasonal or climatological time scales."

- **L360 : check also the previous O'Brien et al. ref which identifies a similar behavior in the convergence as a function of ensemble members.**

**Response:** We thank the reviewer for highlighting the highly relevant work by O'Brien et al. We acknowledge this oversight in our initial literature review and appreciate the reviewer bringing this important reference to our attention. We are pleased to note that O'Brien et al. also identified a similar convergence behavior scaling as $N^{-\frac{1}{2}}$ with ensemble size. In the revised manuscript (Line XX), we have integrated this reference as follows: "The fitting results of Fig. 10b demonstrate that the width of the confidence interval is roughly proportional to $N^{-\frac{1}{2}}$, with the fitting expression being $18.18N^{-\frac{1}{2}} - 1.55$ for this case (see also O'Brien et al. (2011) for similar $N^{-\frac{1}{2}}$ convergence behavior with ensemble size)"."

- **L 365 : As stated earlier, these studies are based on longer model integrations where the temporal average might already dampen the IV effect seen at shorter time scale . In other words perhaps a 10 member ensemble considering multiyear, seasonal means (which includes many events) could be more robust than the author suggest based on 50 members ensemble of 20 day simulations (which each includes a limited numbers of events). When considering multiyear seasonal means, the convergence towards a physical effect in function of ensemble members might be perhaps faster. So less ensemble member required.**

**Response:** We appreciate the reviewer's insightful perspective on the timescale dependence of IV effects. The reviewer rightly points out that temporal averaging over longer integrations (e.g., seasonal or multi-year means) inherently dampens high-frequency IV signals arising from individual synoptic events. This integration effect might indeed allow a smaller ensemble (e.g., 10 members) to converge more rapidly on the forced response in climatological studies, as the averaging process filters out event-scale stochasticity.

To address this point, we have added the following discussion at the end of paragraph: "It is crucial to emphasize that the ensemble size requirements discussed here are specific to the analysis of synoptic-scale processes within this 20-day simulation during the monsoon onset period. Studies focusing on longer-term climatological means (e.g., seasonal averages or multi-year averages) inherently integrate over more weather events. This temporal smoothing might accelerate the convergence towards a robust physical effect in function of ensemble size, which is a promising hypothesis that warrants systematic investigation in future studies. Our findings on the necessity of larger ensembles therefore primarily apply to dust aerosol impacts on synoptic events, where the stochastic component of variability remains dominant and unresolved by temporal averaging."

- **L385. Figure 9. I was wondering how different are the radiative forcings (especially TOA) for E1 and E2.**

**Response:** The difference in dust TOA forcings for E1 and E2 in Figure 9 are shown in Figure R1 below. As seen in Figure R1, the difference in dust-induced TOA forcing between E1 and E2 is minimal. This aligns with the high consistency in dust aerosol distributions across ensemble members, also as demonstrated in Figure 4 of our manuscript. Note that these forcing differences isolate only the direct radiative impact of dust aerosols. They do not account for potential indirect effects or contributions from other variables such as the different cloud cover.

Given the small magnitude of these dust-specific differences in TOA forcing, we elected not to include them in the main manuscript to maintain focus and show them in the supplement materials instead.

We have added these discussions in the revised manuscript as: "To determine whether these contradictory results of precipitation are caused by dust radiative forcings, we also calculate the corresponding dust TOA forcing difference of E1 and E2. The results show that, consistent with the high spatial coherence in dust AOD across ensemble members (Fig. 4), the dust-induced TOA radiative forcing differences between contrasting subsets (e.g., E1 and E2) were found to be very small (Fig. S10)."

[Figure]

**Figure R1. The difference in dust TOA forcing for E1 and E2 in Figure 9.**

- **L395 and Figure 9: About statistical significance: I did not really grasp the method and conclusion here. If you select the E1 and E2 samples to be representative of a type of precipitation response, you automatically increase the statistical significance of the results just due to this preferential sampling, compared to a sample which would contain members with variable type of responses. From this I don't see how to conclude that statistical tests**

*applied to climatic simulations with small ensemble are not meaningful. Maybe I missing something (or a statistical background), that could be explained furthermore.*

**Response:** Thank you for this important question about our statistical significance analysis and conclusions. We would like to clarify both our methodology and the intended interpretation of Figure 9. Our statistical significance testing method employs Student's t-test at each grid cell, comparing 10 samples from the "Control" experiment against 10 corresponding samples from the "Sensitive" experiment to determine if the differences between experiments are statistically significant.

Regarding Figure 9 and our discussion at L395, we recognize that our point may not have been clearly communicated. We are not suggesting that statistical tests applied to climate simulations with small ensembles are meaningless. Rather, we are highlighting a potential pitfall: small ensembles can pass statistical significance tests while still yielding misleading or contradictory results. This occurs because, with a limited number of members (e.g., only 10), the specific random subset of ensemble members chosen for analysis can strongly influence the outcome.

The E1 and E2 combinations were selected specifically to illustrate this issue - they represent two different subsets of ensemble members that, despite showing opposing precipitation responses in certain regions (e.g., the black box area), both achieve statistical significance in these regions. This demonstrates that passing a significance test does not automatically guarantee that the identified signal represents the true physical response. Instead, it could simply reflect the particular random sample of members - If the ensemble size is insufficient, the result may not be robust and could change substantially with a different random selection of members.

To address this concern, we have revised our text to more clearly state: "The statistical significance of the differences is assessed using Student's t-test, performed at each grid cell by comparing 10 samples of ensemble member values from the "Control" experiment against 10 corresponding samples from the "Sensitive" experiment, to determine if the results between the two experiments are significantly different.
…
This analysis demonstrates that achieving statistical significance alone may not guarantee reliable representation of dust impacts when using small ensembles (e.g., only 10 members). Crucially, in practice, the specific subset of 10 members run in a study is essentially a random draw from the larger possible set. It could be any subset, including ones like E1 or E2 that produce statistically significant yet contradictory results. Rather than suggesting statistical tests are not meaningful, our results emphasize the importance of adequate ensemble size to ensure robust characterization of aerosol impacts"

- *L420: Particularly I think when convection is an important component of the meso-scale systems. Orography induced meso-scale system for instance might be less chaotics in term of response to dust.*

**Response:** Thank you for this insightful comment highlighting the important distinction between different types of mesoscale systems. We agree completely that the convective nature of a mesoscale system significantly influences its chaotic behavior and sensitivity to aerosol perturbations.

You raise a good point that orographically-induced mesoscale systems might exhibit less chaotic response to dust compared to systems like the monsoon depression we analyzed. Orographic forcing provides a strong external constraint that can make the system's response more deterministic and potentially less sensitive to initial conditions.

We have revised our manuscript to incorporate this important distinction, adding nuance to our conclusions about ensemble size requirements. Specifically, we've modified the text to clarify that that orographically-forced systems may show more deterministic behavior requiring fewer ensemble members as: "These findings lead to several key conclusions: (1) For mesoscale weather systems, such as monsoon depressions, large ensemble sizes (approximately 30 members) are necessary to obtain robust simulations of dust aerosols effects. (2) For larger-scale processes, such as general monsoon circulation and moisture transport, smaller ensemble sizes (approximately 5 members) may suffice for accurate representation of dust impacts. (3) The chaotic nature of mesoscale systems likely depends on their dominant formation mechanisms—orographically-forced systems (the southern slopes of the Himalayas, for example) might be less sensitive to the initial conditions, which may show greater deterministic behavior. This scale-dependent and process-dependent requirement for ensemble size reflects the inherent predictability differences between synoptic-scale and mesoscale atmospheric processes in dust impacts studies. Please note that our findings of 30 members for mesoscale systems and 5 members for larger-scale processes are specific to our case study of dust effects on the ISM during June 10-30, 2016, and may vary for different aerosol types, regions, or seasons. The optimal ensemble size ultimately depends on the specific research questions, phenomena of interest."

- *L435: and other studies.*

**Response:** Thanks for the reviewer's suggestion. We have added this in the revised text as: "… , which is consistent among most ensemble members and aligns with previous studies (Vinoj et al., 2014; Jin et al., 2015; Lau et al., 2017)."